

# Seasonality of density currents induced by differential cooling

Tomy Doda[1,2], Cintia L. Ramón[1], Hugo N. Ulloa[2], Alfred Wüest[1,2] and Damien Bouffard[1]

[1]Eawag, Swiss Federal Institute of Aquatic Science and Technology, Surface Waters - Research and Management, Kastanienbaum, 6047, Switzerland
[2]Physics of Aquatic Systems Laboratory, École Polytechnique Fédérale de Lausanne, Lausanne, 1015, Switzerland

*Correspondence to*: Tomy Doda (tomy.doda@eawag.ch)

**Abstract.** When lakes experience surface cooling, the shallow littoral region cools faster than the deep pelagic waters. The lateral density gradient resulting from this differential cooling can trigger a cold downslope density current that intrudes at the base of the mixed layer during stratified conditions. This process is known as *thermal siphon* (TS). TS flushes the littoral region and increases water exchange between nearshore and pelagic zones, with possible implications on the lake ecosystem. Past observations of TS in lakes are limited to specific cooling events. Here, we focus on the seasonality of the TS-induced lateral transport and investigate how the seasonally varying forcing conditions control the occurrence and intensity of TS. We base our analysis on one year of observations of TS in Rotsee (Switzerland), a small wind-sheltered temperate lake composed of an elongated shallow region. We demonstrate that TS occurs for more than 50 % of the days from late summer to winter and efficiently flushes the littoral region in ~10 hours. We further quantify the seasonal evolution of the occurrence, intensity and timing of TS. The conditions for the formation of TS are optimal in autumn, when the duration of the cooling phase is longer than the initiation timescale of TS. The decrease in surface cooling by one order of magnitude from summer to winter reduces the lateral transport by a factor of two. We interpret this transport seasonality with scaling relationships relating the daily averaged cross-shore velocity, unit-width discharge and flushing timescale to the surface buoyancy flux, mixed layer depth and lake bathymetry. The timing and duration of the diurnal flushing by TS are associated with the duration of the daily heating and cooling phases. The longer cooling phase in autumn increases the flushing duration and delays the time of maximal flushing, compared to the summer period. Our findings based on scaling arguments can be extended to other aquatic systems to assess, at a global scale, the relevance of TS in lakes and reservoirs.

## 1 Introduction

Lakes are contained by lateral boundaries. These boundaries are characterized by exchanges with the surrounding watershed. The littoral region receives substances from the terrestrial environment including contaminants from human activities (Rao and Schwab, 2007) and it is a zone of accumulation of particulate matter (Cyr, 2017). It is also the place of high biological activity and intensified biogeochemical reactions (Wetzel, 2001). This critical area for the lake ecosystem is not isolated but connected to the pelagic region by cross-shore flows that transport water laterally (MacIntyre and Melack, 1995). The effects



of cross-shore flows on biogeochemistry are often neglected when representing lakes as vertical water columns (Hofmann, 2013) and when monitoring them at the single deepest location (Effler et al., 2010). Yet, cross-shore flows control the residence time of nearshore waters and affect the entire lake ecosystem by redistributing heat, dissolved and particulate compounds across the lake (MacIntyre and Melack, 1995).

Horizontal exchange flows are driven by various mechanisms. The effect of large-scale surface and internal gravity waves on horizontal transport was elucidated already by the fathers of limnology (Forel, 1895; Wedderburn, 1907; Mortimer, 1952). Other types of wind-driven lateral flows result from wind circulation, Ekman transport and differential deepening (Imberger and Patterson, 1989; Brink, 2016). Buoyancy-driven lateral flows can be induced by horizontal gradients of temperature due to differential heating or cooling. Such gradients may occur from spatial heterogeneity of heat fluxes, due to

large-scale differences of meteorological forcing (Verburg et al., 2011), variable geothermal heating (Roget et al., 1993), shading from vegetation (Lövstedt and Bengtsson, 2008), wind sheltering (Schlatter et al., 1997) or differences of turbidity (MacIntyre et al., 2002). Horizontal temperature gradients also occur when waters of varying bathymetry experience a spatially uniform heat flux (Horsch and Stefan, 1988; Monismith et al., 1990). For a given surface area, the volume of water in the shallow littoral region is smaller than its offshore counterpart and will cool or heat faster. The cross-shore circulation resulting

from this bathymetry-induced temperature gradient was called *thermal siphon* by Monismith et al. (1990). The fact that all lakes have sloping boundaries makes thermal siphon an ubiquitous process, yet with a variable importance for horizontal transport.

We restrict this study to the thermal siphon driven by differential cooling, when lake temperature is warmer than the temperature of maximum density (Fig. 1). We will not consider the heating-driven thermal siphon here because the induced

lateral exchange is weaker and it is more sensitive to wind disturbance than the cooling-driven thermal siphon (Monismith et al., 1990; James et al., 1994). Lakes respond to diurnal, synoptic and seasonal changes in the atmospheric forcing. This includes periods of net cooling, with the net surface heat flux directed to the atmosphere (Bouffard and Wüest, 2019). This diel or seasonal surface cooling mixes and deepens the surface layer by penetrative convection. Once the shallow littoral region becomes fully mixed, it cools down faster than the deep pelagic region. Littoral waters become denser and plunge as a

downslope density current, which intrudes at the base of the mixed layer during stratified conditions. At the surface, a reverse flow brings water from the pelagic to the littoral region (Horsch and Stefan, 1988; Monismith et al., 1990). We will hereafter refer to the density current induced by differential cooling simply as *thermal siphon* (TS).

TS has often been mentioned to explain cold anomalies observed in lakes, such as tilted isotherms above the sloping sides (Talling, 1963; Eccles, 1974; MacIntyre and Melack, 1995; Woodward et al., 2017) or cold water intrusions (Wells and

Sherman, 2001; Peeters et al., 2003; Rueda et al., 2007; Forrest et al., 2008; Ambrosetti et al., 2010). However, only a few studies have provided a detailed in situ description of the process. First extensive studies of TS aimed at assessing the fate of heat disposal from power plants in cooling lakes (Adams and Wells, 1984). Warm water discharged into these reservoirs flows onshore at the surface, cools and induces a TS in sidearms. These observations are, however, not representative of naturally formed TS. Cooling lakes constantly receive heat from power plants which increases the intensity and duration of surface





cooling compared to other lakes and reservoirs (Horsch and Stefan, 1988). To the best of our literature review, the diurnal

cycle associated with naturally formed TS was for the first time described from in situ observations by Monismith et al. (1990).

The authors studied a sidearm of Wellington Reservoir (Australia) in summer and described the complete diurnal cycle of

differential heating during daytime and differential cooling during nighttime. They observed cross-shore flows in both cases,

with a velocity of ~2 cm s$^{-1}$. The cross-shore circulation was characterized by inertia, with a delay of several hours between

70 the shift in forcing conditions and the flow reversal. Other studies have reported the presence of TS along the thalweg of

narrow reservoir embayments (James and Barko, 1991a, b; James et al., 1994; Rogowski et al., 2019), but also on the sloping

sides of lakes (Sturman et al., 1999; Thorpe et al., 1999; Fer et al., 2002a, b; Pálmarsson and Schladow, 2008) and between

different basins (Roget et al., 1993). In most of these examples, the characteristics of TS were similar to the description of

Monismith et al. (1990), with a thermal stratification above the sloping bottom, inertial effects and a cross-shore velocity on

the order of 1 cm s$^{-1}$. The work of Fer et al. (2002b) on the sloping sides of Lake Geneva (Switzerland) represents one of the

most extensive studies on TS. The authors collected continuous measurements over two winters and identified density currents

starting in the middle of the night and lasting for ~8 h until the late morning. They measured an increasing discharge with

distance downslope and found that one TS event was able to flush the littoral region almost twice.

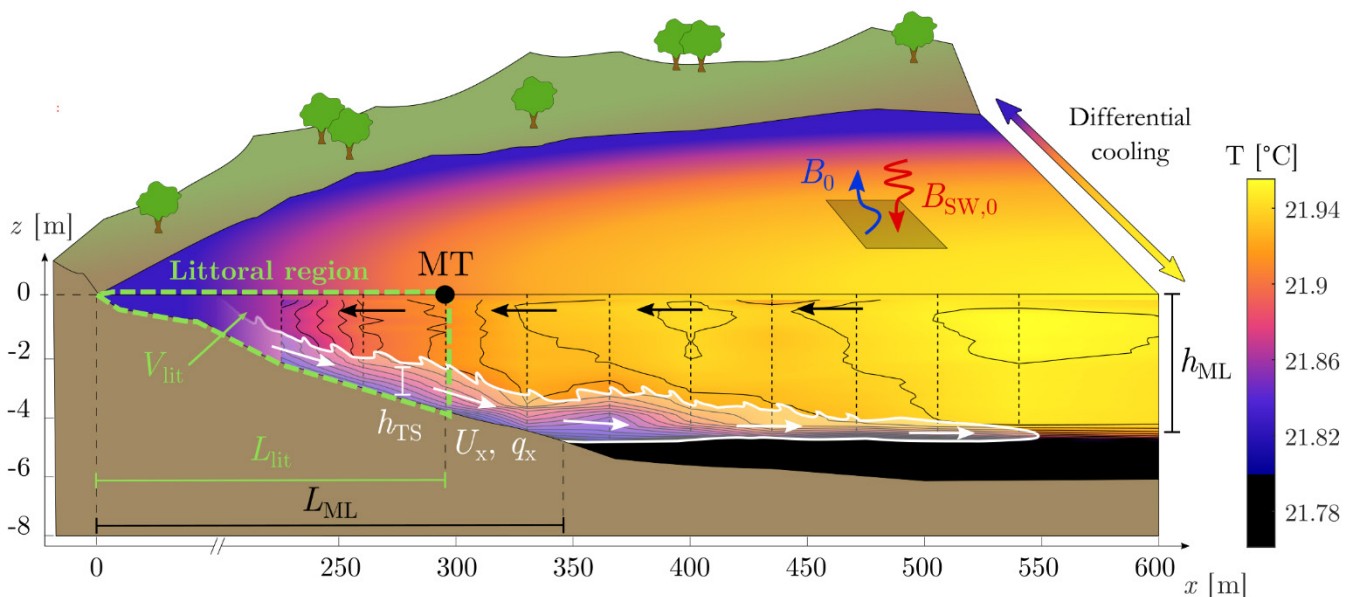

**Figure 1: Data-based schematic of the cooling-driven thermal siphon. The cross-shore temperature field is linearly interpolated from a transect of CTD (Conductivity-Temperature-Depth) profiles collected in the morning on 22 August 2019 (08:20–08:50 UTC). Vertical dashed lines show the location of the profiles. The different variables used for the scaling formulae are represented. Contour lines are 0.01 °C-spaced isotherms. The point MT is the location of velocity measurements.**

The intensity of convective cooling, the duration of the daily cooling period and the lake stratification vary seasonally

(Bouffard and Wüest, 2019). These changes of forcing conditions are expected to affect the occurrence and magnitude of TS

over time. Most of the previous studies, however, focused on specific TS events and did not monitor the process over different



seasons. The three-year-long dataset of Adams and Wells (1984) in Lake Anna (VA, USA) is specific to cooling lakes and does not reflect the seasonality of naturally formed TS. Although the time series collected by Roget et al. (1993) between two lobes of Lake Banyoles (Spain) spans eight months (October-May), the authors did not discuss the seasonal variability of TS.

Lake Banyoles is also a special system where differential cooling is partly due to changes of geothermal fluxes between the lobes, which may modify the seasonality of TS. The six tracer experiments of James and Barko (1991b) were performed between June and September. Their estimated TS velocity varied between the experiments and was higher for days with stronger lateral temperature gradients, but the authors did not identify a clear seasonal trend. They showed from thermistor arrays that the occurrence of differential cooling increased from May to August, suggesting a higher frequency of TS events

in late summer. This assumption was not verified due to the lack of long-term velocity measurements.

To our knowledge, the seasonality of occurrence and intensity of TS has never been investigated in lakes. Empirical relationships linking the forcing and background conditions (surface cooling, stratification) to the cross-shore transport by TS have been derived from dimensional analysis and laboratory experiments (Harashima and Watanabe, 1986), but not verified in the field. Understanding the seasonality of TS allows better predicting its contribution for the exchange between littoral and

pelagic regions of lakes. With this objective in mind, we monitored TS in a small temperate lake over one year. We report here three aspects of the seasonality of TS, related to the occurrence (Sect. 3.3), the intensity (Sect 3.4) and the diurnal dynamics (Sect 3.5) of TS. We further relate this seasonality to the forcing conditions and identify key parameters to predict TS in lakes (Sect. 4). By expanding the past observations of specific TS events to all four seasons, this study provides a comprehensive understanding of the conditions required to form TS in lakes.

## 2 Methods

### 2.1 Study site and field measurements

Rotsee is a small peri-alpine monomictic lake near Lucerne, Switzerland (Fig. 2a). It is located at 419 m above sea level and has a surface area of 0.5 km$^2$ with a mean and maximum depth of 9 m and 16 m, respectively. The main in- and outflows are located at the south-western and north-eastern ends of the lake, respectively. Rotsee is an elongated lake (2.5 km long, 0.2 km

wide), with steep sides (slope of 15°) across its longitudinal axis and more gradual slopes (1.5°) at its two ends. The north-eastern end of the lake finishes with a 200 m long plateau region of approximately 1 m depth that can trigger TS. Moreover, Rotsee is located in a depression, which shelters it from wind. It is thereby famous for international rowing championships and an ideal field-scale laboratory where convective processes are distinctively observable.





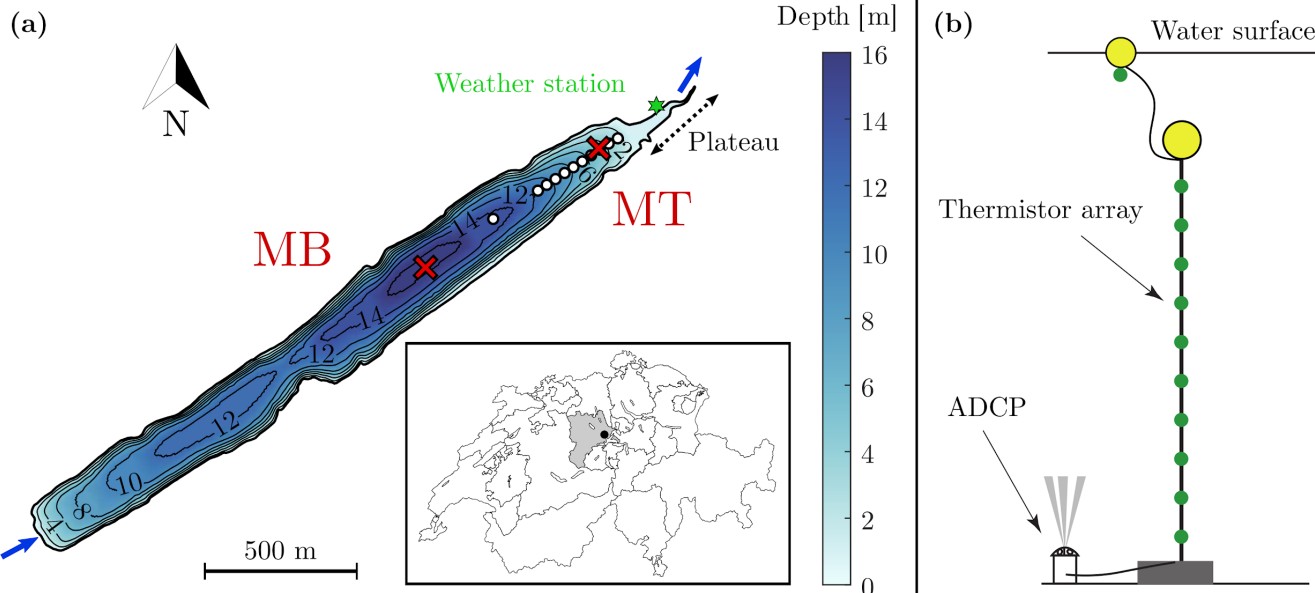

**Figure 2: (a) Bathymetry of Rotsee indicating the location of the two moorings MB and MT (red crosses), the cross-shore transect of CTD profiles (white dots) and the weather station (green star). Blue arrows correspond to the main inflow and outflow. Source: Federal Office of Topography. (b) Schematic of the mooring MT. A similar setup is used for the thermistor array at MB. The detailed setup is provided in Appendix A.**

We monitored the background stratification at the deepest location (mooring MB, approx. 16 m deep) as well as the dynamics of TS at the sloping region (mooring MT, approx. 4 m deep), from March 2019 to March 2020 (Fig. 2a). We briefly describe the moorings below, and provide more detailed information on the specifications and setup of the instruments in Appendix A. Each mooring included an array of thermistors (Fig. 2b). The evolution of the background temperature was monitored at MB with Vemco Minilog II-T loggers (accuracy: 0.1 °C). High-resolution thermistors RBR TR-1050 and RBR Duet TD (accuracy: 0.002 °C) were used at MT to track finer-scale temperature fluctuations near the sloping boundaries. The thermistors were installed from 1 to 13 m above the bottom with 1 m vertical resolution at MB and from 0.25 to 2 m above the bottom with 0.25 m resolution at MT. The evolution of the near-surface temperature (0.2 to 0.3 m below the surface) was monitored with Vemco loggers attached to surface buoys at both locations. An upward-looking Acoustic Doppler Current Profiler (ADCP, Nortek Aquadopp Profiler 1 MHz) was deployed at MT for fine-scale measurements of the bottom currents (Fig. 2b). It collected velocity profiles with a vertical bin resolution of 0.05 m from 0.25 to 3 m above the bottom. Velocity was obtained by averaging 512 burst measurements every 15 min. A week-long maintenance of the moorings was conducted every 40 days on average. The measurement periods by the ADCP were shorter in December and January due to low battery (Appendix A). The monitoring was stopped in May 2019 to avoid any hazard during the European Rowing Championships.

To capture the spatial variability of TS, cross-shore transects of Conductivity-Temperature-Depth (CTD) profiles (Sea&Sun CTD 60M, sampling interval of 0.4 s) were performed during twelve short-term campaigns, between August and December 2019. The chosen CTD lowering speed of 15 cm s$^{-1}$ offered an optimal 6 cm vertical resolution while monitoring



an entire transect with 11 profiles over ~45 min (i.e. short time compared to the duration of flushing by TS, Sect. 3.1). The 11 profiles were distributed along the sloping region of the lake, between 2 and 14 m depth (Fig. 2a). Figure 1 shows a CTD transect collected in the morning on 22 August 2019, as an example.

A WxPRO Campbell weather station was installed at the lake shore in September 2019 (Fig. 2a) and it measured atmospheric pressure, air temperature, wind speed and direction, incoming shortwave radiation and relative humidity every 10 min. In addition, incoming longwave radiation was obtained from the Lucerne weather station (47°02'12'' N, 8°18'04'' E; source MeteoSwiss), ~4 km from Rotsee. The Lucerne station also provided meteorological forcing before September 2019 or to fill data gaps from the reference Rotsee weather station. In these cases, the Lucerne dataset was adjusted based on the Rotsee observations over the measuring period September 2019 to June 2020. Pressure, air temperature and shortwave radiation were

linearly corrected ($R^2 > 0.9$). This simple method did not allow reliable correction to the relative humidity and wind speed and we applied instead a Neural Network Fitting approach to correct these two variables. The shallow neural network consisted of one hidden layer with 50 neurons and was trained by a Levenberg-Marquardt algorithm. 70 % of the reference period September 2019–June 2020 was used to train the neural network and the remaining percentage was equally distributed for validation and testing. All the meteorological variables measured in Lucerne were used as inputs to the network and the

procedure was repeated for each of the two target variables. The network performance was satisfactory, with a coefficient of determination $R^2$ and a root mean square error $E_{RMS}$ between observations and estimates of $R^2 \approx 0.72$ and $E_{RMS} \approx 0.67$ m s$^{-1}$ for wind speed, and $R^2 \approx 0.94$ and $E_{RMS} \approx 4.7\%$ for relative humidity.

## 2.2 Heat and buoyancy fluxes

Heat fluxes were estimated from meteorological data and lake surface temperature at MB. They are defined positive in upward

direction (lake cooling) and negative in the downward direction (lake heating). The non-penetrative surface heat flux $H_{Q_0}$ is defined as

$$H_{Q_0} = H_C + H_E + H_{LW,in} + H_{LW,out} \quad [W \ m^{-2}], \tag{1}$$

where $H_C$ is the sensible and $H_E$ is the latent heat flux, while $H_{LW,in}$ and $H_{LW,out}$ are the incoming and outgoing longwave radiation, respectively. The net surface heat flux $H_{0,net}$ includes the shortwave radiative flux at the surface $H_{SW,0}$:

$$H_{0,net} = H_{Q_0} + H_{SW,0} \quad [W \ m^{-2}]. \tag{2}$$

The different terms of the heat budget are calculated similarly to Fink et al. (2014), except $H_{LW,in}$ which is directly measured by the Lucerne weather station. Cloudiness modulates the proportion of direct and diffuse shortwave radiation and is derived from daily clear-sky irradiance as in Meyers and Dale (1983). We modified the empirical wind and air temperature-based calibration function, $f$, in the sensible and latent heat fluxes estimates of Fink et al. (2014) to take the lake fetch into account,

as proposed in McJannet et al. (2012).

The non-penetrative surface buoyancy flux $B_0$ is inferred from $H_{Q_0}$ under the assumption that only heat fluxes modify the potential energy near the surface (Bouffard and Wüest, 2019):





$$B_0 = \frac{\alpha g H_{Q_0}}{\rho C_{p,w}} \quad [\text{W kg}^{-1}]. \tag{3}$$

In Eq. (3), $\alpha = -(1/\rho)\, \partial\rho/\partial T$ is the thermal expansivity of water [°C$^{-1}$], $g$ is the gravitational acceleration [m s$^{-2}$], $\rho$ is the water density [kg m$^{-3}$] and $C_{p,w}$ is the specific heat of water [J °C$^{-1}$ kg$^{-1}$]. The thermal expansivity was estimated from surface temperature by using the relationship reported in Bouffard and Wüest (2019). To calculate water density, we used the equation of state from Chen and Millero (1986), with a measured average salinity of $S = 0.2$ g kg$^{-1}$ for Rotsee.

Similarly, a radiative surface buoyancy flux $B_{SW,0}$ is obtained as

$$B_{SW,0} = \frac{\alpha g H_{SW,0}}{\rho C_{p,w}} \quad [\text{W kg}^{-1}]. \tag{4}$$

The net buoyancy flux at the surface is $B_{0,net} = B_0 + B_{SW,0}$. $B_{0,net} > 0$ indicates that surface cooling overcomes radiative heating, leading to a net cooling of the water column. $B_{0,net} < 0$ indicates a net heating by shortwave radiation. We use $B_{0,net}$ to identify the daily cooling and heating phases (Sect. 3.1) and we average $B_0$ over the daily cooling phase to quantify the source of convection (Sect. 2.6).

## 2.3 Mixed layer depth

We used hourly averaged temperature data from MB to calculate the thermocline depth ($h_t$) and the mixed layer depth ($h_{ML}$). Different methods for estimating $h_{ML}$ were tested based on the review of Gray et al. (2020) to finally select the temperature-gradient method with a threshold of 0.05 °C m$^{-1}$ for $dT/dz$. The water column was defined as fully mixed when $dT/dz < 0.05$°C m$^{-1}$ at all depths. In this case, $h_{ML}$ was set to the depth of the bottom temperature sensor. For stratified conditions, the thermocline depth was defined as the depth of maximum gradient in $dT/dz$. Starting from $h_t$ and moving upward, the lower end of the mixed layer was reached when the local temperature gradient dropped to $dT/dz < 0.05$ °C m$^{-1}$. When $dT/dz > 0.05$°C m$^{-1}$ at all depths, the entire column was stratified and $h_{ML}$ was zero.

## 2.4 Occurrence of thermal siphons

We used the velocity data from the Aquadopp profiler at MT to estimate the cross-shore transport over the bottom 3 m of the sloping region. Quality control of the 15 min averaged velocity was performed by discarding values with beam correlation below 50 % or signal amplitude weaker than 6 dB above noise floor. The horizontal velocity was projected onto the main axis of the lake (angle of 56° from north) to obtain the cross-shore velocity ($U_x$), defined positive in the offshore direction (x-axis in Fig. 1).

The time series of $U_x$ was then divided into 24 h subsets, starting and ending at 17:00 UTC. Each 24 h subset was analyzed separately to identify TS events. We decided to focus on periods when MT was located in the sloping mixed region, with TS flowing at the bottom of the water column (Fig. 1). This condition allowed us to relate our current measurements to scaling formulae of downslope density currents (Sect. 3.4). A downslope TS event was detected on a specific day if the three following conditions were met:




1. significant cross-shore flow: depth-averaged velocity of the current $\overline{U_x} > 0.5$ cm s$^{-1}$ for at least two hours;

2. weak wind during the selected event: $|L_{MO,avg}|/h_{ML,avg} < 0.5$, with $L_{MO,avg}$ and $h_{ML,avg}$ the averaged Monin-Obukhov length scale (Wüest and Lorke, 2003) and mixed layer depth during the cross-shore flow event, respectively;

3. mixed water column at MT before the onset of the flow: $dT/dz < 0.05$°C m$^{-1}$ at all depths.

Cross-shore flows that did not meet the conditions #2 and #3 were noted as "wind circulation" and "stratified flows", respectively. Further justifications of the above criteria are provided in Appendix C.

## 2.5 Cross-shore transport by thermal siphons

The cross-shore transport was calculated for each identified TS event (Sect. 2.4). We defined for each event sub-periods with continuous positive $U_x$ (Appendix C). We calculated the unit-width volume of water transported over each of these sub-periods:

$$V_{x,i} = \int_{t_{0,i}}^{t_{f,i}} q_x(t)\,dt \quad [\text{m}^2], \tag{5}$$

where $t_{0,i}$ and $t_{f,i}$ are the initial and final times of the sub-period $i$, respectively, and $q_x$ is the unit-width discharge defined as

$$q_x(t) = \int_{z_{bot}(t)}^{z_{top}(t)} U_x(z,t)\,dz \quad [\text{m}^2\,\text{s}^{-1}]. \tag{6}$$

In Eq. (6), $z_{bot}$ and $z_{top}$ are the bottom and top interfaces of the region over which $U_x > 0$ at time $t$. For each day, we then defined the flushing period of TS as the sub-period with the largest volume transported $V_{x,i}$.

Four daily quantities characterizing the cross-shore transport were calculated over the flushing period of each day ($t_0 \leq t \leq t_f$): the average cross-shore velocity $U_{avg}$ and unit-width discharge $q_{avg}$, the maximum cross-shore velocity $U_{max}$ (both in time and depth) and the flushing timescale $\tau_F = V_{lit}/q_{avg}$ with $V_{lit} \approx 500$ m$^2$ the unit-width volume of the littoral region upslope of MT (Fig. 1). $U_{avg}$ and $q_{avg}$ were obtained as follows:

$$U_{avg} = \frac{1}{t_f-t_0} \int_{t_0}^{t_f} \overline{U_x(t)}\,dt \quad [\text{m s}^{-1}], \tag{7}$$

$$q_{avg} = \frac{1}{t_f-t_0} \int_{t_0}^{t_f} q_x(t)\,dt \quad [\text{m}^2\,\text{s}^{-1}]. \tag{8}$$

## 2.6 Scaling formulae

We relate the transport quantities introduced in Sect. 2.5 to the forcing conditions, by using scaling formulae from theoretical and laboratory studies.





**Table 1: Key parameters related to the forcing conditions, bathymetry and TS dynamics. For seasonally varying parameters, the given ranges of values are daily averages.**

| Symbol | Units | Definition and equation | Ranges of values in Rotsee |
|---|---|---|---|
| $\tau_c$ | h | Duration of the cooling phase | $[0,24]$ |
| $\tau_F$ | h | Flushing timescale: $\tau_F = V_{lit}/q_{avg}$ | $[5,20]$ |
| $\tau_{ini}$ | h | initiation timescale: $\tau_{ini} = \tau_{mix} + \tau_t$ | $[2,170]$ |
| $\tau_{mix}$ | h | Mixing timescale: $\tau_{mix} = 0.5(d_{MT}^2 - h_{ML,ini}^2)N^2/B_0(1 + 2A)$ | $[0,250]$ |
| $\tau_t$ | h | Transition timescale: $\tau_t = 2(L_{ML} - l_p)^{2/3}/(1 - d_p/h_{ML})^{1/3}B_0^{1/3}$ | $[2,45]$ |
| $B_0$ | W kg$^{-1}$ | Surface buoyancy flux: $B_0 = \alpha g H_{Q_0}/\rho C_{p,w}$ | $[0,1.4 \times 10^{-7}]$ |
| $B_{0,net}$ | W kg$^{-1}$ | Net surface buoyancy flux: $B_{0,net} = B_0 + B_{SW,0}$ | $[-1.3 \times 10^{-7}, 1.1 \times 10^{-7}]$ |
| $B_{SW,0}$ | W kg$^{-1}$ | Radiative buoyancy flux: $B_{SW,0} = \alpha g H_{SW,0}/\rho C_{p,w}$ | $[-1.8 \times 10^{-7}, 0]$ |
| $d_p$ | m | Depth of the plateau region | 1 |
| $d_{MT}$ | m | Depth at MT | 4.2 |
| $F_G$ | - | Flow geometry parameter: $F_G = \sqrt{<U_x^2>}/\sqrt{<U_z^2>}$ | $[5,18]$ |
| $h_{TS,avg}$ | m | Daily averaged TS thickness: $h_{TS,avg} = q_{avg}/U_{avg}$ | $[1.2,2.4]$ |
| $h_{lit}$ | m | Average depth of the littoral region: $h_{lit} = V_{lit}/L_{lit}$ | 1.7 |
| $h_{ML}$ | m | Mixed layer depth | $[0,15.3]$ |
| $H_{0,net}$ | W m$^{-2}$ | Net surface heat flux: $H_{0,net} = H_{Q_0} + H_{SW,0}$ | $[-200,180]$ |
| $H_{Q_0}$ | W m$^{-2}$ | Non-penetrative surface heat flux | $[-10,250]$ |
| $H_{SW,0}$ | W m$^{-2}$ | Surface shortwave heat flux | $[-300,0]$ |
| $l_p$ | m | Length of the plateau region | 173 |
| $L_{lit}$ | m | Length of the littoral region | 295 |
| $L_{ML}$ | m | Length of the mixed region | $[200,800]$ |
| $q_{avg}$ | m$^2$ s$^{-1}$ | Daily averaged unit-width TS discharge | $[0.005,0.030]$ |
| $q_x$ | m$^2$ s$^{-1}$ | Unit-width TS discharge | - |
| $U_{avg}$ | m s$^{-1}$ | Daily averaged TS cross-shore velocity | $[0.005,0.015]$ |
| $U_{max}$ | m s$^{-1}$ | Daily maximum TS cross-shore velocity | $[0.01,0.04]$ |
| $U_x$ | m s$^{-1}$ | Cross-shore velocity | - |
| $U_z$ | m s$^{-1}$ | Vertical velocity | - |
| $V_{lit}$ | m$^2$ | Unit-width volume of the littoral region | 499 |
| $w_*$ | m s$^{-1}$ | Convective velocity scale: $w_* = (B_0 h_{ML})^{1/3}$ | $[0.001,0.009]$ |

The horizontal velocity scale of TS is $U \sim (B_0 L)^{1/3}$ where $B_0$ is the destabilizing surface buoyancy flux and $L$ is a horizontal length scale (Phillips, 1966). Following Wells and Sherman (2001), we used the horizontal length scale $L_{ML}$ based on the mixed layer depth:

$$U = c_U(B_0 L_{ML})^{1/3} \quad [\text{m s}^{-1}], \tag{9}$$

with $c_U$ a proportionality coefficient. For each value of $h_{ML}$, we calculated $L_{ML}$ as the distance between the north-eastern edge of the lake and the location where the mixed layer intersects the lake sloping bottom (Fig. 1).



The unit-width discharge is scaled by $q \sim (B_0 L)^{1/3} H$, where $H$ is a vertical length scale. We used the average depth of the littoral region upslope of MT ($h_{\mathrm{lit}} = V_{\mathrm{lit}}/L_{\mathrm{lit}}$, Fig. 1) as $H$:

$$q = c_{\mathrm{q}}(B_0 L_{\mathrm{ML}})^{1/3} h_{\mathrm{lit}} \quad [\mathrm{m^2\ s^{-1}}], \tag{10}$$

with $c_{\mathrm{q}}$ a proportionality coefficient. We define the flushing timescale from Eq. (10) as

$$\tau_{\mathrm{F}} = \frac{V_{\mathrm{lit}}}{q} = c_{\mathrm{F}} \frac{L_{\mathrm{lit}}}{(B_0 L_{\mathrm{ML}})^{1/3}} \quad [\mathrm{s}], \tag{11}$$

with $c_{\mathrm{F}} = 1/c_{\mathrm{q}}$. The flushing timescale represents the time that TS needs to flush the littoral region of volume $V_{\mathrm{lit}}$.

In Eqs. (9), (10) and (11), $B_0$ and $L_{\mathrm{ML}}$ were averaged over the cooling period of each day, defined as $B_{0,\mathrm{net}} > 0$. We discuss the choice of the length scales $L_{\mathrm{ML}}$ and $h_{\mathrm{lit}}$ in more detail in Sect. 4.3. The key parameters used for the scaling of the transport and TS dynamics are listed in Table 1.

## 3 Results

### 3.1 Diurnal cycle

The net surface buoyancy flux $B_{0,\mathrm{net}}$ oscillates on a daily timescale with intensity of the diel fluctuation modulated at seasonal timescale (Fig. 3a). We illustrate the diurnal cycle with an example on 9–10 September 2019 (Figs. 3b-d). This summertime period is characterized by a strong day-night variability ideal to elucidate how changes in forcing conditions affect the formation and destruction of TS. The net surface buoyancy flux $B_{0,\mathrm{net}}$ varies from $-2 \times 10^{-7}$ W kg$^{-1}$ during daytime to $1 \times 10^{-7}$ W kg$^{-1}$ during night-time (Fig. 3b). In contrast to $B_{0,\mathrm{net}}$, $B_0$ remains positive over 24 hours, indicating a continuous cooling at the air-water interface. $B_0$ increases during the night, reaches its maximum just before sunrise and decreases again during the day. In late summer, $B_0$ is $\sim 10^{-7}$ W kg$^{-1}$ with limited variability of $\pm 5 \times 10^{-8}$ W kg$^{-1}$. The radiative forcing $B_{\mathrm{SW},0}$ is the main driver of the diurnal variability, with diel variations one order of magnitude larger than $B_0$ during cloud-free summer days like 9–10 September 2019. We observe differential cooling at night, with lateral temperature gradients of $\sim 5 \times 10^{-4}$ °C m$^{-1}$ in late summer (Fig. 1).



**Figure 3: Diurnal cycle of TS. (a) Net surface buoyancy flux as a function of time during the day, represented over one year (from 13 March 2019 to 13 March 2020). DOY stands for "Day of the Year". A 30-day moving average has been applied to smooth the time series. The 45-day-long period without measurement in May 2019 is shown in gray. The contour lines have a spacing of $1 \times 10^{-7}$ W kg$^{-1}$. The dashed line and arrow correspond to the example diurnal cycle shown in (b), (c), (d) (9–10 September 2019; DOY 252–253). (b) Time series of surface buoyancy fluxes defining the cooling and heating phases on 9–10 September 2019. (c) 15 min**
**averaged vertical velocity measured at MT (depth of 4.2 m) as a function of time and height above the sediment. Positive (purple) values correspond to a flow moving upward. Strong vertical movements are the signature of convective plumes. (d) 15 min averaged cross-shore velocity measured at MT as a function of time and height above the sediment. Positive (purple) values correspond to a flow moving offshore. Black lines are 0.02 °C-spaced isotherms. The TS detected by the algorithm and used for the transport calculation is depicted in blue. The vertical resolution of the bottom thermistors is shown by the horizontal black ticks on the right**
**axis (the surface thermistor is not shown).**



We further divide the diurnal cycle into a net cooling phase during the night ($B_0 > |B_{SW,0}|$) and a net heating phase during the day ($B_0 < |B_{SW,0}|$). Starting the period at 17:00 (UTC), the water column is initially thermally stratified by the radiative heating of the daytime period (Fig. 3d). During the first part of the night, convective cooling mixes and deepens the surface layer. The water column becomes entirely mixed at MT around 00:00 (see the vertical isotherms starting around 00:00 in Fig. 3d). The convective plumes intensify in the second part of the night to reach vertical velocities of ~5 mm s$^{-1}$ (Fig. 3c). Three hours after the complete mixing at MT, a cross-shore circulation is initiated, with a downslope flow near the bottom (positive velocity $U_x$ in Fig. 3d) and an opposite flow above (negative velocity $U_x$ in Fig. 3d). The velocity magnitude of ~1 cm s$^{-1}$ is typical for TS. The interface between the two flows oscillates vertically by 1–2 m (~100 % of the TS thickness) suggesting that the transport is limited by the vertical mixing of convective plumes (Fig. 3c). At the end of the cooling phase, the water column is distinctly stratified near the bottom, indicating the presence of persistent colder water flowing downslope. A striking observation is the intensification of TS in phase with the weakening of vertical convection at the onset of radiative heating. The strongest flushing occurs at the beginning of the heating phase and is characterized by a thicker thermal interface between the two opposite flows at ~1.5–2 m above the lake bed (one-third of the total depth at MT). The cross-shore velocity increases above 2 cm s$^{-1}$ and the induced bottom stratification reaches ~0.1 °C m$^{-1}$. This strong flushing lasts for several hours during the net heating phase, while radiative heating increases and re-stratifies the water above the density current. The cross-shore flow is weakening in the late morning, yet continues until noon.

The example detailed above presents all characteristics of typical TS, with different phases and inertia between the cooling period and the flushing. Other examples, provided in Appendix B, show that the timing and magnitude of the flow change with the forcing conditions.

### 3.2 Seasonal variability of the forcing conditions

Based on the transport scaling (Sect. 2.6), two key parameters for the formation of TS are $B_0$ and $h_{ML}$, which is related to $B_{0,net}$. Hereafter, we refer to these parameters as "forcing conditions" of TS. The daily intensity of convective cooling is estimated based on the convective velocity scale $w_* = (B_0 h_{ML})^{1/3}$, where $B_0$ and $h_{ML}$ are the daily averaged surface buoyancy flux and mixed layer depth during the cooling phase at MB. These two parameters are both seasonally dependent and their relationship is characterised by an annual hysteresis cycle (Fig. 4a). Although the surface heat flux $H_{Q_0}$ increases from winter to summer, the strong seasonal variability of $B_0$ (one order of magnitude larger in summer than in winter) is mostly due to seasonal changes in the thermal expansivity $\alpha$.

From spring to summer, the mixed layer remains shallow (< 2 m) and $B_0$ increases from ~$10^{-9}$ W kg$^{-1}$ in March to ~$10^{-7}$ W kg$^{-1}$ in August. The latter results in an increase of the convective velocity scale from $w_* \approx 3.8 \pm 0.7$ mm s$^{-1}$ to $w_* \approx 6.7 \pm 0.9$ mm s$^{-1}$ over the same period (Figs. 4a, b). The lake undergoes net daily heating over this period ($B_{0,net} < 0$),





with a cooling phase that remains shorter than 15 h (Fig. 4b). In August, $B_{0,net} \approx 0$ and daily heating is balanced by daily cooling. At this period, the intensity of convective mixing reaches its maximum.

**Figure 4: Seasonal variability of the forcing conditions over one year (from March 2019 to March 2020). (a) Daily averages of mixed layer depth as a function of surface buoyancy flux during cooling periods. Horizontal and vertical error bars represent the standard deviation of $B_0$ and $h_{ML}$, respectively. Error bars are emphasized for DOY 77, 262 and 313. Gray dashed lines indicate the corresponding convective velocity scale. The black dashed line with arrows is a qualitative representation of the annual cycle. (b) Monthly averages of convective velocity scale $w_*$ during the cooling phase (green line), duration of the cooling phase $\tau_c$ (blue line) and net surface buoyancy flux $B_{0,net}$ (linearly interpolated colormap). Error bars represent the standard deviation of $w_*$ and $\tau_c$.**

From September to January, the lake undergoes every day a net cooling ($B_{0,net} > 0$) and the mixed layer deepens by convection with an average rate of ~0.1 m d$^{-1}$, leading to a complete mixing in mid-December. Over the same period, $B_0$ continuously decreases due to the drop in $\alpha$ with colder temperatures. This decrease in $B_0$ balances the deepening of $h_{ML}$ and $w_*$ remains constant around $w_* \approx 7 \pm 1$ mm s$^{-1}$ from August to November. The duration of the cooling phase increases in autumn to reach its maximum in November ($\tau_c \approx 21 \pm 2$ h). Convective cooling is occurring almost continuously at that time. In winter, the convective velocity $w_*$ drops to $w_* \approx 3.5 \pm 1$ mm s$^{-1}$ and the effect of the cooling phase is again balanced



by the heating phase ($B_{0,\text{net}} \approx 0$, $\tau_c \approx 15 \pm 3$ h in February). Note that winter 2019–2020 was warm and Rotsee remained mixed from December to February (no inverse stratification, surface temperature $T > 4°C$).

The error bars in Fig. 4a indicate the diurnal variability of surface cooling that reaches a maximum in summer. For the three selected examples (black error bars), the daily standard deviation of $B_0$ is around $2.6 \times 10^{-8}$ W kg$^{-1}$ in late September (DOY 262, pink dot), against $0.9 \times 10^{-8}$ W kg$^{-1}$ in November (DOY 313, purple dot) and $0.3 \times 10^{-8}$ W kg$^{-1}$ in March (DOY 77, green dot). The differences between the cooling and heating phases of the diurnal cycle (Fig. 3) are thus reduced in autumn, with a more constant surface cooling and a weaker radiative heating ($B_{0,\text{net}} > 0$) than in summer.

## 3.3 Seasonal occurrence of thermal siphons

We apply our detection algorithm for cross-shore flows (Sect. 2.4) on 227 days with continuous measurements at MT from March 2019 to February 2020. We identify 156 days with a significant cross-shore flow (69 % of the days with observations) over which 85 days (37 %) were reported as TS events. The remaining days with identified cross-shore flow were either associated with wind circulation for 13 days (56 %) or stratified flows for 58 days (26 %). We estimate the percentage of occurrence of TS relative to the number of days with measurements ($p_{\text{TS}}$) for each month (Fig. 5). Before late July, the mixed layer is usually shallower than 4 m (Fig. 4a) and stratified flows prevail at MT (Fig. 5). The first events identified as TS occur on 28 July and 31 July ($p_{\text{TS}} = 12.5$ % in July). Yet, it is rather in August that TS reaches a significant percentage of occurrence of ~50 %. The occurrence increases in autumn to reach 87 % in November. In winter, TS is less common: it occurs for only 38 % of the days in December, two days in early January (not shown in Fig. 5 due to the few days measured during this month) and is not observed later in winter.

The 15-days moving average in Fig. 5 reveals the short-term variability due to synoptic changes in the meteorological conditions that naturally modulate the seasonal pattern. Although the monthly averaged occurrence increases from August to November, the bi-weekly percentage of occurrence drops over shorter periods. These periods are associated with strong heating that re-stratifies the water column at MT, reduces surface cooling at night and prevents the formation of TS. We also notice that the 15 d averaged occurrence reaches 100 % in November, corresponding to a period where TS is present over 25 consecutive days.

Although TS becomes more frequent in autumn, the percentage of occurrence of cross-shore flows ($p_{\text{CS}}$) remains constant around 80–90 % from July to November. Lateral flows observed at MT in spring and summer primarily result from direct wind forcing or subsequent stratified mixing. These flows may also include TS intruding before reaching MT and thereby not counted as downslope TS (further details in Appendix C). These different processes become less frequent in autumn due to the deeper mixed layer and are replaced by downslope TS. Overall, there seems to be an effect of TS on the number of cross-shore flows if we compare the period from July to December ($p_{\text{CS}} \approx 80 \pm 14$ %) with the rest of the year ($p_{\text{CS}} \approx 47 \pm 12$ %), suggesting that TS is the main mechanism connecting the littoral and pelagic regions.



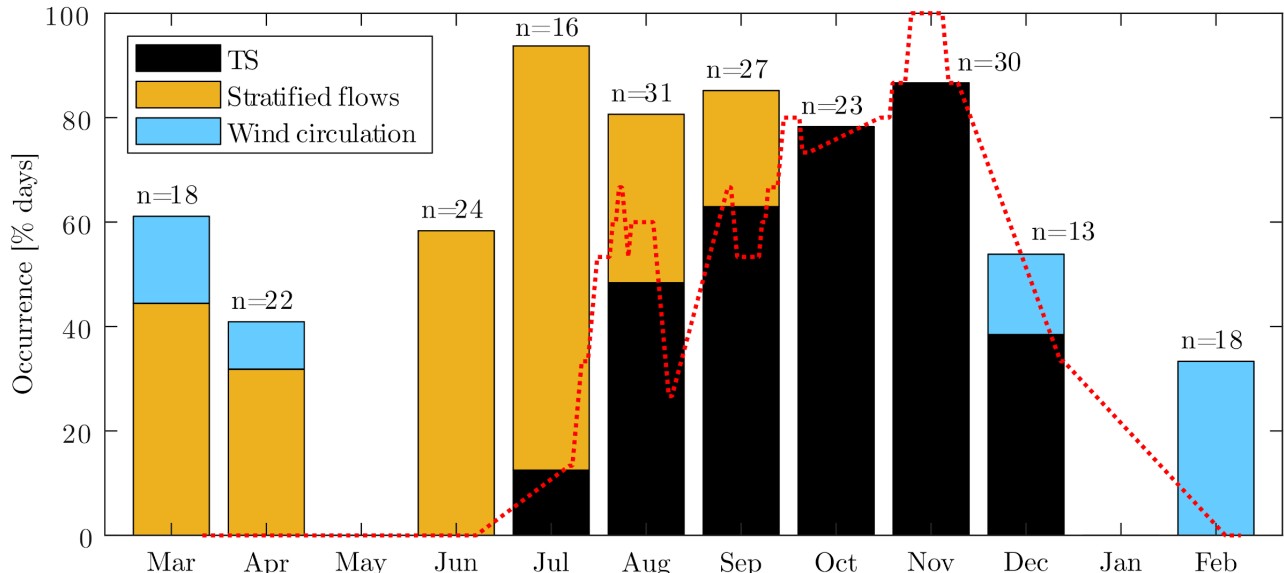

**Figure 5: Monthly occurrence of TS and other cross-shore flows at MT, expressed as a percentage of the days with measurements.** The other cross-shore flows are divided between wind circulation ($|L_{MO}|/h_{ML} > 0.5$) and stratified flows at MT ($dT/dz > 0.05$°C m$^{-1}$). The red dashed line indicates the 15-day moving average of the percentage of occurrence of TS. The number of days with measurements (n) is shown for each month. Months with less than 10 days of measurements have been removed (May 2019 and January 2020).

**3.4 Scaling the cross-shore transport**

We compare our field-based transport estimates for the TS events identified in Fig. 5 with the laboratory-based scaling formulae introduced in Sect. 2.6 (Fig. 6). The daily average and maximum cross-shore velocities are linearly correlated with the horizontal velocity scale from Eq. (9) ($R^2 > 0.5, p_{val,F} < 0.01$; Figs. 6a, b). The daily average velocity $U_{avg} \approx 0.9 \pm 0.2$ cm s$^{-1}$, three times lower than the daily maximum velocity $U_{max} \approx 2.6 \pm 0.7$ cm s$^{-1}$ that occurs in the morning (Fig. 3). The best linear fits are given by $U_{avg} = 0.33 \cdot (B_0 L_{ML})^{1/3}$ and $U_{max} = 0.99 \cdot (B_0 L_{ML})^{1/3}$. Despite the natural variability, a seasonal trend is distinguishable, with a decrease of $U_{avg}$ and $U_{max}$ by a factor of two from July to December. This is consistent with the weakening of convective cooling observed in Fig. 4.

The unit-width discharge $q_{avg}$ and flushing timescale $\tau_F$ are also well predicted by the scaling formulae (10) and (11), despite a larger scatter than for the cross-shore velocity ($R^2 \approx 0.2 - 0.3$, Figs. 6c, d). The unit-width discharge is $q_{avg} \approx 0.015 \pm 0.004$ m$^2$ s$^{-1}$, corresponding to an average thickness of $h_{TS,avg} = q_{avg}/U_{avg} \approx 1.8 \pm 0.2$ m and a flushing timescale of $\tau_F \approx 9.7 \pm 2.7$ h. The best relationships are given by: $q_{avg} = 0.34 \cdot (B_0 L_{ML})^{1/3} h_{lit}$ ($R^2 = 0.27$) and $\tau_F = 2.99 \cdot L_{lit}/(B_0 L_{ML})^{1/3}$ ($R^2 = 0.18$). The strong daily variability is explained by the fluctuating thickness of TS during the convective period (Figs. 3c, d), which affects the calculation of the discharge (further details in Appendix C). As observed for





the cross-shore velocity, the cross-shore transport weakens from summer to winter: $q_{\mathrm{avg}}$ decreases and $\tau_{\mathrm{F}}$ increases by a factor of two. The flushing timescale is $\tau_{\mathrm{F}} \approx 7$ h in summer but reaches $\tau_{\mathrm{F}} \approx 20$ h in winter.

Figure 6: (a) Daily averaged cross-shore velocity, (b) daily maximum cross-shore velocity, (c) daily averaged unit-width discharge and (d) flushing timescale as a function of scaling formulae (Sect. 2.6). The equation of the linear regressions, the coefficient of determination ($R^2$) and the p-value of an F-test ($p_{\mathrm{val,F}}$) are indicated.

**3.5 Flushing period**

The diurnal cycle described in Sect. 3.1 varies at seasonal scale, as the forcing conditions change. This is especially important for the period over which TS flows, hereafter called the flushing period (Sect. 2.5). To assess the changes in the flushing period, we average the diurnal cycle monthly between August and December (Fig. 7). The cooling and heating phases are illustrated with the diurnal cycle of $B_{0,\mathrm{net}}$. The duration of the cooling phase (blue shadow in Fig. 7) markedly increases from August to December, as already observed in Fig. 4b. In December, the heating phase (red shadow) takes place only from 10:00



to 14:00 on average, whereas it lasts from 07:00 to 17:00 in August. The histograms of $|B_{0,net}^{max}|$ show that the magnitude and variability of $B_{0,net}$ decrease over the same period.

These changes of the forcing conditions have a direct effect on the cross-shore transport. The unit-width discharge in Fig. 7 (white dotted line) is obtained by monthly averaging the 24 h time series $q_x(t)$. The discharge $q_x(t)$ is calculated from Eq. (6) during the flushing period $t_0 \leq t \leq t_f$ and $q_x(t) = 0$ for $t < t_0$ or $t > t_f$. In August and September, $q_x$ remains weak in the afternoon and at the beginning of the cooling phase. It increases at night to reach its maximum in the morning with the beginning of the heating phase. The flow generally stops in the late morning, depicted by a drop of $q_x$ around 10:00 in Fig. 7a. This cycle corresponds to the example of Fig. 3. The daily peak of flushing is also present in October and November but it is more spread with a continuous increase of cross-shore transport at night, starting in the evening already (Figs. 7e, g). The average flushing duration of TS is thus larger. Periods with a net surface cooling over several days are common in November and are associated with TS flowing continuously for more than a day (example in Appendix B). In December, the peak of discharge is reduced and the cross-shore transport is nearly continuous over 24 h (Fig. 7i). The time of maximal $q_x$ (vertical black dashed line in Fig. 7) is delayed from August to December. We associate this maximal $q_x$ to the heating phase as it always occurs ~1–2 h after $B_{0,net}$ becomes negative (except in December where the peak is less pronounced).

The diurnal cycle of the cross-shore transport is also described by the flow geometry parameter introduced by Ulloa et al. (2021) as the ratio between the root mean square (RMS) of the cross-shore velocity and the RMS of the vertical velocity:

$$F_G(t) = \frac{\sqrt{<U_x(t)^2>}}{\sqrt{<U_z(t)^2>}},$$ (12)

where $<\ldots>$ denotes a depth-average. $F_G$ informs on the relative magnitude of TS with respect to the surface convection. We expect $F_G$ to be close to unity when surface convection is the only process acting on the water column. A deviation from unity towards larger values of $F_G$ (i.e., $F_G \gg 1$) implies that the flow is anisotropic, with a stronger cross-shore component, as TS develops. A sharp peak of the monthly averaged $F_G$ is visible at the beginning of the heating phase, from August to November (Fig. 7). It matches the time of maximal $q_x$ and results from the combined effect of TS-induced flushing (increase of $U_x$) and convection weakening (decrease of $U_z$).





**Figure 7: Seasonal variability of the diurnal cycle. (a), (c), (e), (g), (i) Monthly average of the diurnal cycle of the unit-width discharge**
**$q_x$ and $F_G$ parameter. Only days with observed TS between August and December have been averaged. The shaded white area**
**corresponds to average discharge ± standard deviation. The monthly averaged net buoyancy flux represented as a colormap is**
**normalized by the maximum absolute value of each month. The vertical black dashed line indicates the time of maximal $q_x$. The**
**transition timescale $\tau_t$ defines the onset time represented by a pink vertical dashed line. (b), (d), (f), (h), (j) Histograms of the**
**maximum absolute value of the net buoyancy flux for each month. The red vertical line corresponds to the maximum value $|B_{0,\text{net}}^{\max}|$**
**of the averaged diurnal cycles used to normalize $B_{0,\text{net}}$ in (a), (c), (e), (g), (i).**

To give a theoretical estimation of the onset of the flushing, we calculate the monthly averaged transition timescale $\tau_t$
from Ulloa et al. (2021) as:

$$\tau_t = \frac{2\left(L_{\text{ML}} - l_p\right)^{2/3}}{B_0^{1/3}\left(1 - d_p/h_{\text{ML}}\right)^{1/3}} \quad [\text{s}], \tag{13}$$





with $l_p \approx 170$ m and $d_p \approx 1$ m for the length and depth of the plateau region of Rotsee, respectively. This timescale is based

on a three-way momentum balance between the lateral pressure gradient due to differential cooling and the inertial terms. It

represents the time needed to create TS under constant surface cooling, which is a reasonable assumption for night-time cooling

in Rotsee (Fig. 3). The onset time depicted by a vertical pink dashed line in Fig. 7 corresponds to $(t_{sunset} + \tau_t)$, with $t_{sunset}$

the time of sunset when $B_{SW,0}$ drops to zero. The increase of $\tau_t$ from $\tau_t \approx 3$ h in late summer to $\tau_t \approx 20$ h in winter leads to a

later theoretical onset of TS in winter, despite the earlier time of sunset. Based on this seasonal variability of $\tau_t$, TS is expected

to start in the evening in summer but in the morning in winter. This theoretical onset, however, can be improved by considering

the initial stratification at the beginning of the cooling phase, as further discussed in Sect. 4.4.

## 4 Discussion

### 4.1 Seasonality of thermal siphons

From our one-year-long measurements, we concluded that both the occurrence and intensity of TS vary seasonally in lakes

with shallow littoral zones comparable to Rotsee. TS occurs regularly from late summer to early winter, with a maximum

frequency in November, and is absent the rest of the year (Sect. 3.3). While the frequency of TS increases in autumn, the

intensity of the net cross-shore transport decreases compared to the summer period (Sect. 3.4). The diurnal cycle is well defined

in summer and divided into a cooling phase at night and a heating phase during daytime (Sect. 3.1). TS is initiated during the

second part of the night and lasts until late morning, with a maximal flushing at the beginning of the heating phase. In autumn,

the flushing duration increases and the time of maximal flushing is shifted later in the day (Sect 3.5).

To explain the seasonality of TS, we need to relate it to the seasonal variability of the forcing conditions. The duration

of the cooling phase ($\tau_c$) is a key parameter to understand the occurrence of TS and the flushing period, while the intensity of

surface cooling ($B_0$) and the stratification ($h_{ML}$) parametrize the cross-shore transport. We discuss the effects of the forcing

conditions on the seasonality of TS in the following sections.

### 4.2 Effects of the forcing conditions on the occurrence of thermal siphons

The occurrence of TS can be predicted by estimating an initiation timescale ($\tau_{ini}$) and comparing it with the duration of the

cooling phase ($\tau_c$). $\tau_{ini}$ represents the minimum cooling duration required to form TS. TS will occur if $\tau_c > \tau_{ini}$. In initially

stratified surface waters, $\tau_{ini}$ results from two timescales: $\tau_{ini} = \tau_{mix} + \tau_t$, with $\tau_{mix}$ the time needed to vertically mix the

littoral region. To estimate $\tau_{mix}$, we use the deepening rate of the mixed layer expressed as (Zilitinkevič, 1991):

$$\frac{dh_{ML}}{dt} = (1 + 2A) \frac{B_0}{h_{ML}N^2} \quad [\text{m s}^{-1}], \tag{14}$$

where $N^2 = -(g/\bar{\rho})(d\rho/dz)$ [s$^{-2}$] is the squared buoyancy frequency below the mixed layer, $\bar{\rho}$ is the depth-averaged water

density and $A \approx 0.2$ is an empirical coefficient. The model of Eq. (14) assumes that the mixed layer deepens by convection

only, without any wind contribution. This assumption is valid for calm conditions, which prevail in Rotsee due to wind




sheltering (Zimmermann et al., 2019). The average duration required for mixing the water column at MT, assuming a constant
surface cooling $B_0$ and an initial mixed layer depth $h_{\mathrm{ML,ini}}$, can be derived from Eq. (14) as:

$$\tau_{\mathrm{mix}} = \left(d_{\mathrm{MT}}^2 - h_{\mathrm{ML,ini}}^2\right)\frac{N^2}{2B_0(1+2A)} \quad [\mathrm{s}]. \tag{15}$$

In Eq. (15), the buoyancy frequency squared is approximated as $N^2 = -(g/\bar{\rho})[\rho(z = -h_{\mathrm{ML,ini}}) - \rho(z = -d_{\mathrm{MT}})]/[d_{MT} - h_{\mathrm{ML,ini}}]$. The depth $d_{\mathrm{MT}}$ corresponds to the maximum depth of the littoral region upslope of MT (Fig. 1), which has to be mixed
to observe downslope TS at MT. Once the littoral region is fully mixed, the onset of TS is not immediate. We observe a delay
that we interpret as $\tau_t$, in agreement with previous laboratory and numerical studies (Finnigan and Ivey, 1999; Ulloa et al.,
2021).

From the seasonality of the forcing conditions (Fig. 4), we predict the optimal period for the occurrence of TS based on
$\tau_{\mathrm{ini}}$ (Fig. 8a). In summer, the shallow mixed layer ($h_{\mathrm{ML}} \approx 2$ m) and strong nighttime surface cooling ($B_0 \approx 10^{-7}$ W kg$^{-1}$) lead
to a short transition timescale of $\tau_t \approx 2 \pm 0.5$ h. However, the strong stratification near the surface limits the deepening of
the mixed layer and causes $\tau_{\mathrm{mix}}$ to be too large for the water column to mix at MT (orange period in Fig. 8a). The littoral
region remains stratified for most of the nights ($h_{\mathrm{ML}} < d_{\mathrm{MT}}$) and the occurrence of downslope TS at MT is low ($\tau_{\mathrm{ini}} > \tau_c$ on
average). Downslope TS starts to be observed in late summer, when the mixed layer reaches $d_{\mathrm{MT}}$ before the end of the night.
Starting in October, the monthly averaged mixed layer is deeper than $d_{\mathrm{MT}}$ and $\tau_{\mathrm{mix}} \approx 0$. At the same time, $\tau_t$ increases because
of the deepening of the mixed layer and the decrease of $B_0$. The transition timescale becomes longer than $\tau_c$ in winter (purple
period in Fig. 8a), reaching its maximum of $\tau_t \approx 23 \pm 7$ h in January. This prevents TS to form in winter, except for days
with continuous surface cooling (i.e., $\tau_c > 24$ h). The transition timescale decreases in spring but $\tau_{\mathrm{mix}}$ increases
simultaneously due to the re-stratification of the littoral region, which prevents TS to occur. The conditions to observe
downslope TS at MT are thus optimal when $h_{\mathrm{ML}} > d_{\mathrm{MT}}$ and $\tau_t \ll \tau_c$, which is the case between September and December
(gray shaded period in Fig. 8a). This period coincides well with our frequent observations of TS in autumn. The limiting factor
for the formation of TS in spring and summer is $\tau_{\mathrm{mix}}$ and it is $\tau_t$ in winter.

In addition to the seasonality, the biweekly averaged occurrence shows variability at a shorter timescale (Fig. 5). This
indicates that a change of forcing conditions ($B_{0,\mathrm{net}}$) over short periods may enhance or prevent the formation of TS, by
modifying the occurrence condition $\tau_{\mathrm{ini}} < \tau_c$. A cold and cloudy period ($B_{0,\mathrm{net}} \approx 0$ during the heating phase, $B_{0,\mathrm{net}} \gg 0$
during the cooling phase) is associated with a weak stratification, a short $\tau_{\mathrm{ini}}$ and can lead to $\tau_{\mathrm{ini}} < \tau_c$, even during the summer
stratified period. Conversely, a warm and sunny period ($B_{0,\mathrm{net}} \ll 0$ during the heating phase, $B_{0,\mathrm{net}} \approx 0$ during the cooling
phase) can re-stratify the water column, increase $\tau_{\mathrm{ini}}$ and prevent TS to occur. The magnitude of $B_{0,\mathrm{net}}$ during the following
day can also modify the duration of the cooling phase ($\tau_c$). A cloud-free and warm day ($B_{0,\mathrm{net}} \ll 0$) can stop the cooling phase
earlier than a cloudy and cold day ($B_{0,\mathrm{net}} \approx 0$). To include these different effects on the occurrence of TS, we average the net
buoyancy flux over 48 h ($B_{0,\mathrm{net}}^{2\mathrm{d}}$), for each 24 h subset starting at 17:00 UTC. $B_{0,\mathrm{net}}^{2\mathrm{d}}$ is the average of $B_{0,\mathrm{net}}$ between 00:00
UTC on the first day of the 24 h subset and 23:59 UTC on the following day. We compare the values of $B_{0,\mathrm{net}}^{2\mathrm{d}}$ between days





with and without TS (Fig. 8b). We notice that, as expected, days with TS are characterized by higher $B_{0,net}^{2d}$ than days without

TS, which is confirmed by a t-test at a 5 % significance level, from August to December.

**Figure 8: Effects of seasonal changes of forcing conditions on the occurrence of TS. (a) Time series of monthly averages of the cooling**
**duration ($\tau_C$), the transition and mixing timescales ($\tau_t$, $\tau_{mix}$) and the depth of the mixed layer with respect to the depth at MT ($h_{ML}'$).**
**Note the log-scale for the axis of timescales. The shaded areas ($\tau_C$, $\tau_t$, $\tau_{mix}$) and error bars ($h_{ML}'$) indicate the monthly standard**
**deviation. The gray shaded period corresponds to optimal conditions for the occurrence of TS. (b) Box plots of the two-day-averaged**
**net surface buoyancy flux for each month, depending on the occurrence of TS.**

Other mechanisms affect the occurrence of TS on a daily basis, such as wind events. Direct wind forcing either enhances

or suppresses TS. We identified several wind events from September to December that prevented any cross-shore flow in the

offshore direction (i.e., no significant flow detected by the algorithm) or generated a strong cross-shore flow that was reported

as wind circulation in Fig. 5.





Although the occurrence of TS over short periods may vary from one year to another, we expect that the seasonal trend previously described would be repeated every year in Rotsee. We also suggest that a similar trend should be common in other temperate lakes. Most of the TS events reported for other systems were observed in late summer or autumn, corresponding to the occurrence period in Rotsee (Monismith et al., 1990; James and Barko, 1991a, b; James et al., 1994; Rogowski et al., 2019). James and Barko (1991b) showed that differential cooling in Eau Galle Reservoir (WI, USA) already developed in early summer, but during cold front events only. For their study system, the percentage of days with differential cooling increased from 31 % in May to 95 % in August. However, these high values cannot be interpreted as the percentage of occurrence of TS since differential cooling can happen without necessarily forming a cross-shore circulation. Pálmarsson and Schladow (2008) also found that differential cooling was rare in May and June in Clear Lake (CA, USA). Only two nights were cold enough to drive cooling-driven TS. In winter, TS events are not frequent in Rotsee but they were reported for larger systems such as Lake Geneva in January (Thorpe et al., 1999; Fer et al., 2002a, b). A possible explanation is that $\tau_t$ depends on the bathymetry (Eq. 13) and can be lower than $\tau_c$ even in winter, if $d_p/h_{ML}$ is small or the slope is high (shorter $L_{ML}$ for the same $h_{ML}$). In addition, small lakes like Rotsee cool faster and reach the temperature of maximum density before larger and deeper lakes, which stops convective cooling earlier in winter.

### 4.3 Effects of the forcing conditions on cross-shore transport

The seasonality of the cross-shore transport induced by TS can be examined by using the scalings for the cross-shore velocity $U_x$ (Eq. (9)) and the unit-width discharge $q_x$ (Eq. (10)). The timescale $\tau_F$ is also a key parameter to assess the seasonality of the littoral region flushing (Eq. (11)). We showed that these scalings reproduce the observed seasonality in Rotsee (Fig. 6). The 95 % confidence interval of each proportionality coefficient is $0.32 \leq c_U \leq 0.34$ for $U_{avg}$, $0.95 \leq c_U \leq 1.02$ for $U_{max}$, $0.33 \leq c_q \leq 0.36$ for $q_{avg}$ and $2.83 \leq c_F \leq 3.15$ for $\tau_F$. The daily averaged and maximum velocities of TS in Rotsee decrease from summer to winter. This is explained by the decrease of $B_0$ by a factor 10, which overcomes the increase of $L_{ML}$ due to the deepening of the mixed layer (Fig. 4). As a result of decreasing $U_{avg}$, the unit-width discharge also drops over the occurrence period, leading to a longer flushing timescale in winter ($\tau_F \approx 15$ h) than in summer ($\tau_F \approx 7$ h). Yet, $\tau_F$ remains shorter than the duration of the cooling phase (Fig. 8a), which demonstrates that TS is efficient to transport littoral waters offshore. In summer, the duration of a TS event is around 10 h (Fig. 3) and the entire region upslope of MT ($V_{lit} \approx 500$ m$^2$) can be flushed by a single event. For autumn days with longer cooling periods, TS can flow continuously over the day and flush the littoral region more than twice in 24 h. The use of the length scale $h_{lit}$ in Eq. (10), which is constant over time, implies that the daily averaged thickness of TS $h_{TS,avg} = q_{avg}/U_{avg}$ should not show any seasonal variability. This is what we observe in Rotsee: $h_{TS,avg}$ remains close to its yearly average of $1.8 \pm 0.2$ m, without any seasonal trend.

We further compare our scalings for $U_x$, $q_{avg}$ and $\tau_F$ with previous studies on buoyancy-driven flows. The velocity scale $U \sim (B_0 L)^{1/3}$ from Phillips (1966) is commonly used, with $L$ defined as the horizontal length scale along which a lateral density gradient is set. This length scale varies with the experimental configuration and basin geometry (Table 2). In laboratory





experiments with a localized buoyancy loss, $L$ is the length along which the destabilizing buoyancy flux applies (Harashima and Watanabe, 1986; Sturman et al., 1996; Sturman and Ivey, 1998). For more realistic systems undergoing uniform surface cooling, $L$ is the length of the littoral region, which can be defined as the plateau zone (Ulloa et al., 2021) or the region vertically mixed by convection (Wells and Sherman, 2001). In the first case, $L = l_p$ is a constant whereas $L = L_{ML}$ varies with the stratification in the second case. We chose to use $L_{ML}$ because it corresponds to the region that is affected by differential

cooling. We expect that a deeper mixed layer will increase the temperature difference $\Delta T$ between the littoral and pelagic regions, resulting in a higher velocity $U_x$. In Eqs. (10) and (11), the transport is expressed as a function of the velocity scale and the size of the littoral region of length $L_{lit}$ and depth $h_{lit}$ (Fig. 1). We defined the littoral region based on the location of MT because the discharge that we measured corresponds to the flow out of this region. The vertical length scale $h_{lit}$ can also be chosen as the depth of the plateau ($d_p$) (Sturman and Ivey, 1998; Wells and Sherman, 2001; Fer et al., 2002b; Ulloa et al.,

2021). However, we think that defining it based on the location where $q_x$ is measured is consistent with the fact that $q_x$ varies spatially, increasing with distance from the shore (Fer et al., 2002b). Using $d_p$ is more appropriate to predict the discharge from the plateau region as in Ulloa et al. (2021). Despite these different choices of length scales, the coefficient $c_q \approx 0.34$ is close to other estimates from the literature (Table 2). Harashima and Watanabe (1986) found that $c_q$ is increasing with the flux Reynolds number defined as $Re_f = B_0^{1/3} h_{lit}^2 / L_{ML}^{2/3} \nu$, with $\nu \approx 1.5 \times 10^{-6}$ m$^2$ s$^{-1}$ the kinematic viscosity of water. The

empirical relationship $c_q \approx 0.43 - 1.67/Re_f^{1/2}$ was obtained for $Re_f > 50$. For the days where TS is observed in Rotsee, $Re_f$ varies from 250 in summer to 50 in winter, with an average value of ~140 which leads to $c_q \approx 0.29$, close to our estimate of $c_q \approx 0.34$.

**Table 2: Comparison of the transport scaling formula $q = c_q h (B_0 L)^{1/3}$ between different studies on sloping basins. The unit-width discharge $q$ is measured at the location $x_q$ along the cross-shore x-axis. The plateau region has a length $l_p$ and a depth $d_p$. $L_{ML}$ is the**
535 **length of the mixed region of depth $h \leq h_{ML}$ and $l_{forc}$ is the length over which the destabilizing forcing $B_0$ applies. The length $\overline{h(x < x_q)}$ is the average depth of the region upslope of $x_q$.**

| Study | Geometry of the basin | Location $x_q$ | Horizontal length scale $L$ | Vertical length scale $h$ | Coefficient $c_q$ |
|---|---|---|---|---|---|
| Harashima and Watanabe (1986) | Plateau/Infinite slope | $x_q < l_p$ | $L = l_{forc} = l_p$ | $h = d_p$ | $0.13 \leq c_q \leq 0.33$ |
| Sturman and Ivey (1998) | Plateau/Slope | $x_q = l_p$ | $L = l_{forc} = l_p$ | $h = d_p$ | 0.2 |
| Ulloa et al. (2021) | Plateau/Slope | $x_q = l_p$ | $L = l_p$ | $h = d_p$ | 0.35 |
| This study | Plateau/Slope | $x_q \leq L_{ML}$ | $L = L_{ML}$ | $h = \overline{h(x < x_q)}$ | 0.34 |


### 4.4 Effects of the forcing conditions on the flushing period

The seasonality of the forcing conditions does not only affect the occurrence and magnitude of TS but also the diurnal dynamics of the littoral flushing (Fig. 7). The onset of TS always occurs several hours after the beginning of the cooling phase. This

delay is modulated by $\tau_{\mathrm{ini}}$ (Sect. 4.2) and consistent with previous studies where the onset of TS was observed at night (Monismith et al., 1990; James et al., 1994; Fer et al., 2002b; Pálmarsson and Schladow, 2008) or in the morning (Rogowski et al., 2019; Sturman et al., 1999). $\tau_{\mathrm{t}}$ is a good estimate for the onset time when the water column is initially mixed ($\tau_{\mathrm{mix}} = 0$). In summer and warm autumn days, however, the onset of TS occurs later than predicted by $\tau_{\mathrm{t}}$ only, due to the initial stratification at the beginning of the cooling phase ($\tau_{\mathrm{mix}} \neq 0$). This leads to a delay in the increase of $q_{\mathrm{x}}$ and $F_{\mathrm{G}}$ (Fig. 7a, c).

In autumn, TS events lasting for more than 24 h lead to monthly averaged $q_{\mathrm{x}} > 0$ before the onset time predicted by $\tau_{\mathrm{t}}$ (Fig. 7e, g). In winter, $\tau_{\mathrm{t}} \rightarrow 24$ h implies that the rare TS events are all characterized by a continuous flow (Fig. 7i), without any diurnal cycle of flushing anymore.

When a diurnal cycle is present in summer and autumn, the flushing period ends several hours after the beginning of the heating phase (Fig. 3, Fig. 7). This inertia was already reported by Monismith et al. (1990), who observed TS until the mid-

afternoon, despite the opposite pressure gradients due to differential heating. In addition to the delay between the end of the cooling and the end of the flow, we also showed that a peak of flushing occurred a few hours after the beginning of the heating phase (Fig. 3). The time of maximal flushing is shifted later in the day from summer to winter, as the heating phase starts later (Fig. 7). The maximal flushing appears related to the change in the forcing, that is the transition from the cooling phase to the heating phase. This finding suggests that convective plumes eroding the flow control the weak initial transport during the

cooling phase. A weakening of convection reduces this vertical mixing and finally enables the flushing to reach its maximum intensity. Further investigations are required to understand this process better.

### 4.5 Practical recommendations to predict and measure thermal siphons in other lakes

Rotsee is an ideal field-scale laboratory to investigate TS, due to wind sheltering and its elongated shape minimizing complex recirculation. To assess the seasonality of TS in other lakes, long-term velocity and temperature measurements in the sloping

region are required. The developed algorithm to detect TS events (Sect. 2.4) and calculate the cross-shore transport (Sect. 2.5) fulfilled our requirements but remained lake specific. The limitations of the algorithm are discussed in more detail in Appendix C. Further development is needed to build a robust algorithm with lake independent physically based criteria that could be applied to other systems.

The effects of a different bathymetry can be predicted from the scaling discussed in the previous sections. A shallower

nearshore plateau region or a steeper sloping region would decrease the transition timescale $\tau_{\mathrm{t}}$ (Eq. (13)) causing the onset of TS to happen earlier and more often. Higher slopes would also decrease the length of the mixed region $L_{\mathrm{ML}}$ and reduce the horizontal velocity of TS (Eq. (9)). Past observations of TS reported horizontal velocities ranging from ~0.1 cm s$^{-1}$ (James and Barko, 1991a, b) to ~10 cm s$^{-1}$ (Roget et al., 1993; Fer et al., 2002b), with $U_{\mathrm{x}} \sim 1$ cm s$^{-1}$ in most cases (Monismith et al., 1990;



Sturman et al., 1999; Pálmarsson and Schladow, 2008; Rogowski et al., 2019). The cross-shore transport $q_x$ and flushing
timescale $\tau_F$ are strongly dependent on the size and depth of the littoral region considered (Eqs. (10) and (11)). A deeper littoral
region (larger $h_{lit}$) would lead to a stronger discharge and a longer littoral region (larger $L_{lit}$) would take more time to flush.
The increase of discharge with depth was reported in Lake Geneva by Fer et al. (2002b). Values of TS thickness and discharge
can be one order of magnitude larger in deeper lakes than Rotsee (Thorpe et al., 1999; Fer et al., 2002a; Rogowski et al., 2019).

Information on lake bathymetry and thermal structure are needed to predict the occurrence ($\tau_{ini} < \tau_c$) and intensity of
TS ($U_x, q_x, \tau_F$) under specific forcing. This approach does not consider wind effects, which deserve further investigations.
Windier conditions would hinder TS by locally enhancing and reducing the cross-shore transport. In this case, the scalings (9),
(10) and (11) should be modified to take wind shear into account. The littoral region would also be mixed faster under windy
conditions and the effects of wind mixing on $\tau_{mix}$ should be included in Eq. (15).

## 5 Conclusions

The flushing of the littoral region by cross-shore flows increases the exchange between nearshore and pelagic waters. In this
study, we investigated one of the processes that enhance the renewal of littoral waters, the so-called thermal siphon (TS) driven
by differential cooling. From a one-year-long monitoring of TS in a small temperate wind-sheltered lake, we quantified the
seasonality of the cross-shore transport induced by TS. This seasonality is related to changes of the intensity of surface cooling
(surface buoyancy flux $B_0$) and the lake stratification (mixed layer depth $h_{ML}$). Three aspects of the seasonality of TS are
highlighted. First, TS is a recurring process from late summer to winter (when $\tau_c > \tau_{ini}$), occurring on ~80 % of the autumn
days. Second, the seasonal changes in the TS-induced transport are well reproduced by the scalings $U_{avg} \approx 0.33 \cdot (B_0 L_{ML})^{1/3}$,
$q_{avg} \approx 0.34 \cdot (B_0 L_{ML})^{1/3} h_{lit}$ and $\tau_F = V_{lit}/q_{avg} \approx 2.99 \cdot L_{lit}/(B_0 L_{ML})^{1/3}$, with $L_{ML}$ the length of the region mixed by
convection and $V_{lit} = L_{lit} h_{lit}$ the unit-width volume of the littoral region. This study provides a robust field validation of
laboratory and theoretically based scalings. Third, the TS-induced flushing period follows the seasonal changes of the cooling
and heating phases, by evolving from a well-defined diurnal cycle in summer to a more continuous flow in winter.

Our results demonstrate that TS significantly contributes to the flushing of the nearshore waters. This process occurs frequently
during the cooling season (> 50 % of the time), each time flushing the entire littoral region. We stress that this buoyancy-
driven transport is, perhaps counterintuitively, stronger in summer and in the morning. Such shift in the timing should help
revisit the role of TS for the transport of dissolved compounds, with, for instance, stronger exchange between littoral and
pelagic waters at a time of high primary production (summer and daytime). Overall, this study provides a solid framework to
integrate the role of TS in the lake ecosystem dynamics.



## Appendix A: Instrumentation

**Table A1: Specifications and setup of the sensors from the two moorings and the meteorological station. For the ADCP, only the complete days of measurements used for the transport estimates are taken into account in the measurement periods.**

| Sensors | Accuracy | Resolution | Number | Setup | Periods of measurements |
|---|---|---|---|---|---|
| *Thermistor array at MB* | | | | | |
| Vemco Minilog II-T (temperature) | 0.1 °C | 0.01 °C | 14 | Sampling interval [s]: 120; Approximate depths [m]: 0.2, 3, 4, 5, 6, 7, 8, 9, 10, 11, 12, 13, 14, 15 | . 13.03.19–24.04.19 . 07.06.19–02.07.19 . 16.07.29–03.09.19 |
| RBR Duet TD (temperature, pressure) | T: 0.002 °C; P: 0.05 % | T: 5×10⁻⁵ °C; P: 0.001 % | 1 | Sampling interval [s]: 1; Approximate depth [m]: 15 | . 05.09.19–18.10.19 . 24.10.19–16.12.19 . 20.12.19–29.01.20 . 12.02.20–05.06.20 |
| *Thermistor array at MT* | | | | | |
| Vemco Minilog II-T (temperature) | 0.1 °C | 0.01 °C | 1 | Sampling interval [s]: 120; Approximate depth [m]: 0.2 | |
| RBR TR-1050 (temperature) | 0.002 °C | 5×10⁻⁵ °C | 8 | Sampling interval [s]: 10; Approximate depths [m]: 2.4, 2.7, 2.9, 3.2, 3.4, 3.7, 3.9, 4.2 | Same periods than above |
| RBR Duet TD (temperature, pressure) | T: 0.002 °C; P: 0.05 % | T: 5×10⁻⁵ °C; P: 0.001 % | 1 | Sampling interval [s]: 1; Approximate depth [m]: 3.4 | |
| *ADCP at MT* | | | | | |
| Nortek Aquadopp Profiler 1 MHz (velocity) | 1 % | 0.1 mm s⁻¹ | 1 | Cell size [m]: 0.05; Burst interval [s]: 900; Sampling interval [s]: 0.5; Number of samples per burst: 512 | . 13.03.19–23.04.19 . 07.06.19–01.07.19 . 16.07.29–02.09.19 . 05.09.19–16.10.19 . 24.10.19–03.12.19 . 21.12.19–07.01.20 . 12.02.20–03.03.20 |
| *Campbell meteostation WxPRO* | | | | | |
| Young Wind Sentry (wind speed and direction) | Speed: 0.5 m s⁻¹; Direction: 5° | Speed: 0.001 m s⁻¹; Direction: 0.1° | 1 | Sampling interval [s]: 600; Height [m]: 3 | |
| Sensirion SHT75 (air temperature and relative humidity) | T: 0.4 °C; RH: 4 % | T: 0.01 °C; RH: 0.03 % | 1 | Sampling interval [s]: 600; Height [m]: 2 | 14.09.19–05.06.20 |
| Setraceram 278 (barometric pressure) | 1 hPa | 0.01 hPa | 1 | Sampling interval [s]: 600; Height [m]: 2 | |
| Apogee Instruments SP-110-SS pyranometer (solar radiation) | 5 % | 1 W m⁻² | 1 | Sampling interval [s]: 600; Height [m]: 3 | |



## Appendix B: Examples of thermal siphon events with different durations

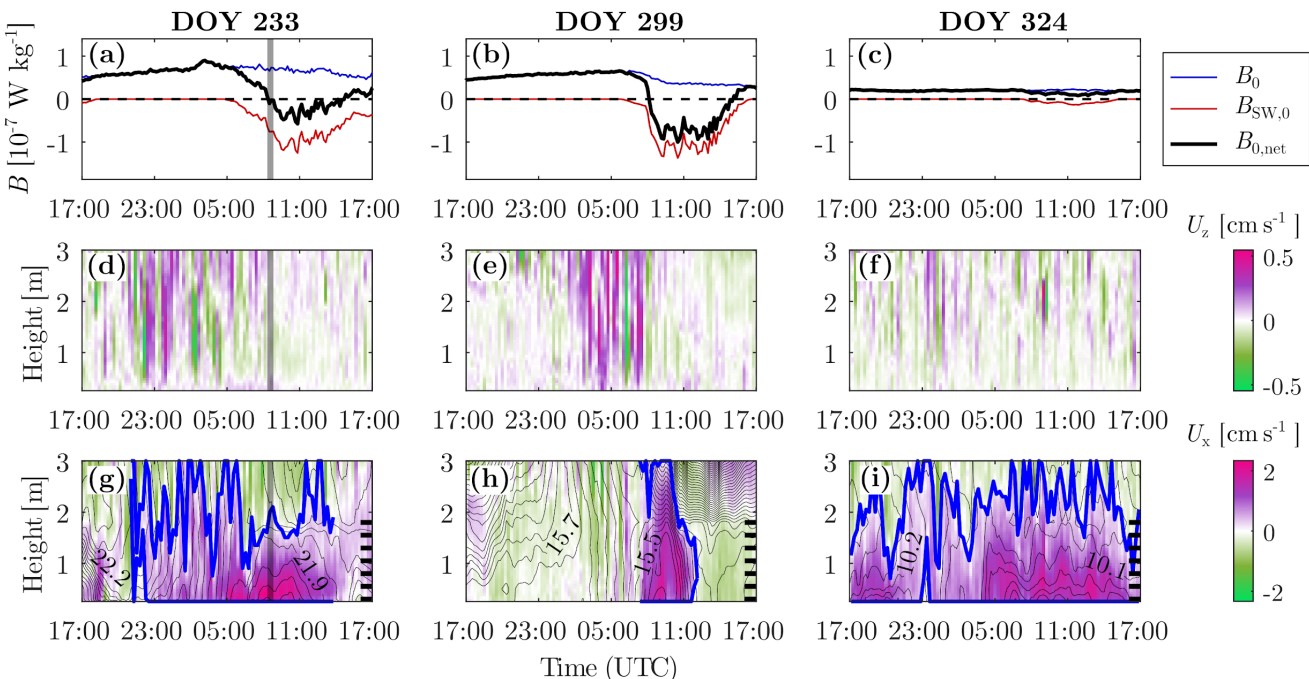

**Figure B1: Three examples of TS events, with (a)-(c) buoyancy fluxes, (d)-(f) vertical velocity and (g)-(i) cross-shore velocity, as in Fig. 3. (a), (d), (g) Long TS event on 21–22 August 2019 due to short heating phases on both days. (b), (e), (h) Short TS event on 26–27 October 2019 due to strong heating on both days (re-stratification). (c), (f), (i) Continuous flushing on 20–21 November 2019 due to continuous net cooling. See the caption of Fig. 3 for more details about each subpanel. The shaded area in (a), (d), (g) indicates the time of the CTD transect shown in Fig. 1.**

## Appendix C: Identification of thermal siphons by the algorithm

The developed algorithm used to detect TS events (Sect. 2.4) and calculate the cross-shore transport (Sect 2.5) aimed at automatizing the identification of TS and thereby at limiting subjective bias in the characterization of the process. We manually assessed the performance of the algorithm over different days. We further tested the validity of the identified TS events with the fact that a cross-shore transport resulting from TS should be associated with a decrease of water temperature (Fig. 1). We calculated the correlation $r(U_x, T)$ between the cross-shore velocity $U_x$ and the temperature $T$ from the thermistor array at MT, at each depth and during the cross-shore flow events. The negative correlation during 93 % of the identified TS events ($r(U_x, T) < -0.48$ according to the 95 % confidence interval from t-test) and the lack of clear negative correlation (at a 5 % significance level) during the other days justified the skills of the detection algorithm. Yet, the algorithm has limitations regarding reproducibility and should be modified for studying TS in another system.

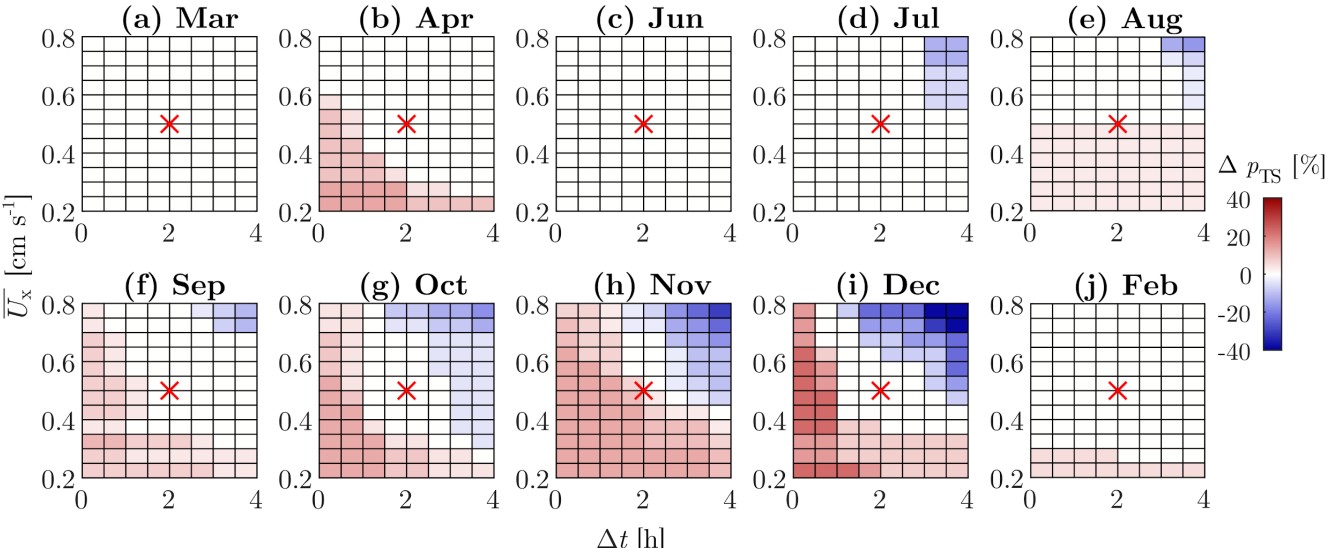

**Figure C1: Sensitivity analysis for the criteria used to define cross-shore flows. The effects of modifying the depth-averaged cross-shore velocity $\overline{U_x}$ and the averaging duration $\Delta t$ are shown as changes in the percentage of occurrence of TS ($p_{TS}$) for each month. $\Delta p_{TS}$ is the percentage difference with respect to the occurrence of Fig. 5, obtained with $\overline{U_x} = 0.5$ cm s⁻¹ and $\Delta t = 2$h (reference point shown by the red cross).**

The definition of a significant cross-shore flow by the algorithm (i.e., $\overline{U_x} > 0.5$ cm s⁻¹ for at least two hours) is lake-specific. This criterion is based on typical examples of TS in Rotsee, where $\overline{U_x}$ is above 0.5 cm s⁻¹ over most of the flushing period. Theoretical estimates of the period between the onset time and the end of the cooling phase (e.g., $\tau_c - \tau_{ini} \approx 10$ h in autumn) show that the majority of TS events last more than two hours and justify the chosen threshold for the duration. From a sensitivity analysis (Fig. C1), we notice that the occurrence of TS does not change significantly if the duration threshold is increased by a few hours. Shorter cross-shore flows events must be discarded because they are not consistent with the gravitational adjustment triggered by differential cooling. They could be wind-driven or resulting from free surface convection. In Rotsee, cross-shore flows driven by internal waves are expected to be short and to be discarded by the algorithm ($T_{V1H1}/2 \approx$ 2.5 h in summer with $T_{V1H1}$ the period of V1H1 internal waves). Other types of data could be included to detect TS, such as lateral temperature gradients, near surface velocity (return flow) or vertical velocity (convective plumes). The use of other techniques, like machine learning algorithms, could also be a useful approach to better identify TS events.

The filter used to discard days with wind circulation is based on the criterion $|L_{MO}|/h_{ML} = \kappa^{-1}|u_*/w_*|^3 > 0.5$, with $\kappa = 0.41$ the von Kármán constant and $u_*$ the friction velocity. The ratio $u_*/w_*$ expresses the relative importance of wind shear compared to convection. The threshold value of 0.5 was successfully tested on wind circulation events observed in Rotsee in winter. This filter discards the strongest wind events but cross-shore flows with $|L_{MO}|/h_{ML} < 0.5$ can still be affected by a wind peak during the flushing period (e.g., interaction between wind and TS). Being more conservative and decreasing the threshold value from 0.5 to 0.1 would discard TS events mainly in December (when $w_*$ is low) but would not significantly modify the occurrence of TS during the other periods (Fig. C2). In lakes more affected by wind, additional filters could be





implemented based on observed oscillations of the thermocline (e.g., wavelet analysis), estimates of the period of internal waves and identification of upwelling events (e.g., Wedderburn and Lake numbers) (Imberger and Patterson, 1989).

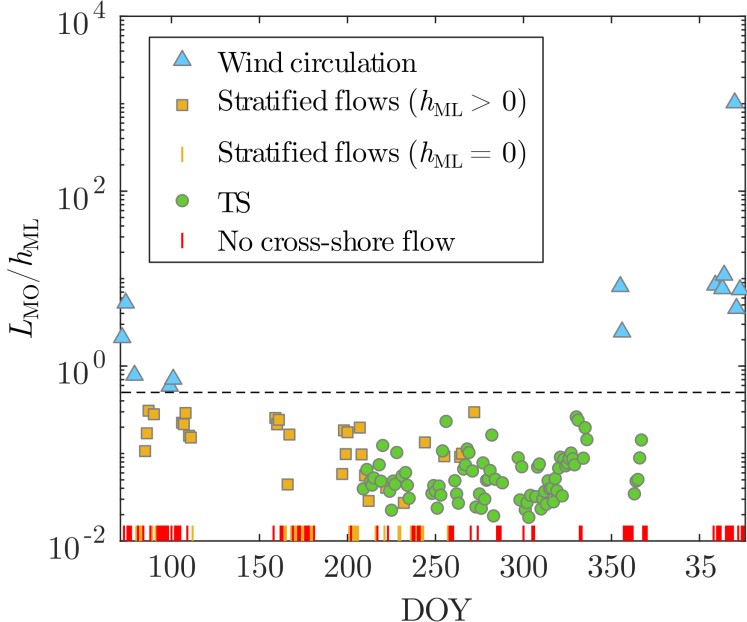

**Figure C2: Filtering of the 227 days analyzed, shown as a time series of $L_{MO}/h_{ML}$ from Mars 2019 to Mars 2020. Days without a significant cross-shore flow or without a mixed layer are depicted as bottom ticks. Stratified flows are cross-shore flows without complete mixing at MT. The wind circulation threshold $L_{MO}/h_{ML} = 0.5$ is shown by the horizontal dashed line.**

By discarding days with stratified conditions at MT, the second filter does not consider a TS that intrudes before reaching MT and appears as an interflow. The occurrence of TS in summer is thereby underestimated. In Fig. 5, the higher percentage of occurrence of stratified flows in July (81 %) compared to spring (around 40–50 %) suggests a possible contribution of intruding TS. From visual inspection of each event, we estimated that intruding TS could represent almost half of the stratified flows in July. Including downslope TS only was justified by the fact that the transport properties of interflows can be different from downslope gravity currents, for which the scaling formulae were derived. In addition, intruding TS events could be confounded with other baroclinic flows and further filtering steps would have been required to correctly detect them.

The transport quantities of TS are averaged over the flushing period (Sect. 2.5). The onset of the flushing is challenging to define because of the interaction between TS and convective plumes (Fig. 3c). The cross-shore velocity field can be very variable over time, with vertical fluctuations of the region with positive $U_x$ (Fig. 3d). We decided to include only the period over which the region with positive $U_x$ remained at the same depths (i.e., the limits of the flushing period correspond either to vertical displacements of the region of positive $U_x$ or to $U_x \leq 0$ at all depths). This approach might sometimes include only a part of the TS event in the transport calculations. Daily averaged velocities are not expected to be affected but the average thickness and discharge vary between days, depending if the vertical oscillations of the TS interface are included or not. We

think that this is a possible reason for the larger variability observed for $q_{avg}$ and $\tau_F$ (Fig. 6). A possibility to reduce this variability would be to apply a moving average on the velocity data.

**Code and data availability**

The raw data, processed data and the data displayed in the figures are available for download, along with the scripts used for all the analysis (temporary link[1]: https://drive.switch.ch/index.php/s/9Dh6hbYZatdTzPr).

**Author contribution**

TD and DB designed the field experiments. TD led the data collection and analysis, with the help of DB, HNU and CLR. TD wrote the initial draft of the manuscript and all co-authors commented and edited the text.

**Competing interests**

The authors declare that they have no conflict of interest.

**Acknowledgements**

We would like to sincerely thank our technician Michael Plüss for organizing the field campaigns and helping setting up and maintaining the different instruments. We are grateful to the Canton of Luzern, the municipalities of Luzern and Ebikon, the
Rowing Centre Lucerne-Rotsee, the associations Quartierverein Maihof and ProNatura, and the Rotsee-Badi for their support in our measurements. We are also indebted to Bieito Fernández-Castro for his help in estimating the heat fluxes. We also thank Love Råman Vinnå, Edgar Hédouin, Josquin Dami and Alois Zwyssig for their assistance in the field. Discussions with Mathew Wells, Oscar Sepúlveda Steiner and Love Råman Vinnå helped to improve the data analysis and the quality of the manuscript. The meteorological data from the Lucerne weather station has been provided by MeteoSwiss, the Swiss Federal
Office of Meteorology and Climatology.

**Financial support**

This study was financed by the Swiss National Science Foundation (project "Buoyancy driven nearshore transport in lakes", HYPOlimnetic THErmal SIphonS, HYPOTHESIS, grant no. 175919).

---

[1] This link is available for the reviewers. After the review process, the final version of the data and the scripts will be uploaded to the ERIC-open database (https://opendata.eawag.ch/) and linked to a DOI number.





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
