# Peer review of "Seasonality of density currents induced by differential cooling"

_Hydrology and Earth System Sciences, 2021_

## Author Comment (AC1)

**Interactive discussion on "Seasonality of density currents induced by differential cooling"**

(Tomy Doda, Cintia L. Ramón, Hugo N. Ulloa, Alfred Wüest, and Damien Bouffard)

**Response to Referee #1**

**R (Referee):** *This paper describes an experimental work aimed at studying the occurrence of thermal siphons in Rotsee, a shallow lake sheltered by the wind. "Thermal syphon" indicates a physical process driven by differential cooling mainly due to bathymetry, which has important ecological implications, enhancing the hydraulic exchange between the littoral and the pelagic zone. The Authors made use of 1-year velocity and temperature data in the shallower area to detach thermal syphons. They focused on the frequency of occurrence over the year and analysed the forcing data suitable to explain this seasonality.*

*They developed a state-of-art experimental work, winding the situ descriptions of the phenomena, which are not so frequent in the literature, especially when aimed at investigating the process over the seasons.*

**A (Authors):** We thank Referee #1 for the critical assessment of the manuscript. His/her comments about the methods and the presentation of the results will improve the clarity of the paper. We addressed them below.

**RC1 (Referee's Comment #1):** *The main limits of this contribution are the weak readability of the paper and the case-specific algorithm proposed to analyse the phenomena. With regard to the first aspect, my suggestion is to be much more concise in the text and in the figures, limiting the number of information to the most relevant ones, or alternative to help the reader to distinguish between the more and less relevant.*

**AR1 (Authors Response #1):**

- Regarding the readability of the paper, we will follow the reviewer's suggestion to make the text more concise by removing unnecessary information and moving details to the Appendix. We will also send the manuscript for English language editing, which should help to gain clarity. The figures will be improved, as explained in more detail in AR8 and AR9.
- Regarding the algorithm used to detect thermal siphon (TS) events, we acknowledge that it is lake-specific, as several criteria are based on threshold values that can vary between systems. Specificity of the algorithm has already been discussed in Sect. 4.5 (lines 560-563) and in Appendix C. However, such an algorithm can serve as a basis for detecting TS in other lakes. Hence, the study is not "case-specific" and the general structure of the algorithm (Sect. 2.4) can

readily be used in other systems. The only changes consist in adapting the threshold values and possibly modifying the filters to distinguish TS from other cross-shore flows. We will update Appendix C to better reflect this question. The lines 616-617 will now read: *"The general structure of the algorithm can be used in other systems. Yet, several criteria are lake-specific and must be adapted to the system of interest. We discuss the limitations of the algorithm below and provide suggestions of improvement."*

We also suggest adding the following sentence on line 562: *"The general structure of the algorithm (Sect. 2.4) can serve as a basis for detecting TS in lakes, by adapting the lake-specific criteria to other systems."*

**RC2:** *More specific suggestions are listed in the followings.*

*Methods 2.1. The computation of the wind speed doesn't seem satisfactory for two reasons: 1- the location of the meteorological station is hardly well representative of the wind conditions over the lake's surface 2 – the methodology to derive wind data from the Lucerne station is not properly justified. Given the sheltering of this lake, a correlation between these sites' data is unlike and I am quite doubtful about the suitability of a neural network algorithm to estimate it, given the local character on the wind field. Despite I don't know the typical wind speeds at these site, I believe that in relative terms an error $E_{RMS}$ of 0.67 m/s is high. On the contrary, the approach is valuable for the other variables. I think that these sources of uncertainty must be accounted and discussed. Actually I do not have any suggestion regarding the solution of problem 1, apart for discussing this limitation in case of absence of other suitable data. With regard to problem 2, instead, I believe that is necessary to introduce an uncertainty in the fluxes evaluation.*

**AR2:**

- Regarding problem 1, we acknowledge that the wind speed can be different between the location of the meteorological station and the lake center (MB), although the distance between the two points is less than one kilometer. However, we think that our measurements are representative of the wind conditions over the nearshore plateau region, where thermal siphons are created (i.e., north-eastern end of the lake). The wind speed cannot only vary between the meteorological station and the lake center, but also all over the lake's surface. The spatial variability of the wind speed is a common problem for the in-situ estimation of heat fluxes in lakes, as it cannot be resolved by a single meteorological station. In the case of Rotsee, we do not expect a significant effect of the spatial variability of wind speed on the daily averaged heat and buoyancy fluxes, because of the small size of the lake and the prevalence of low wind conditions. The daily-averaged wind speed is indeed less than 1 m s$^{-1}$ for 80 % of the days and less than 2 m s$^{-1}$ for 95 % of the days. We propose to add a sentence about the assumption of

spatial homogeneity on lines 154-155: "*We assume that the meteorological conditions and the heat fluxes are spatially uniform over the lake's surface (0.5 km$^2$).*"

- Regarding problem 2, we agree that estimating the wind velocity from the Luzern station introduces uncertainties, which we quantified by the root mean square error on lines 150-152. However, we believe that a Neural Network (NN) approach is the most robust method to correct the Lucerne data to Rotsee. The performance of NN for estimating the spatial variability of wind speed has been demonstrated by Philippopoulos and Deligiorgi (2012). For other examples of studies using this approach, we refer to our answer AR6 to Referee #2.

To illustrate the performance of NN in Rotsee, we compare the estimated wind speed from NN with the measured wind speed in Lucerne and Rotsee stations over a month (Fig. R1.1). The period shown in Fig. R1.1 is not part of the NN training period. Although wind speed is larger in Lucerne than in Rotsee, a coherent correlation between the two sites is observed for most of the wind events. The NN approach reproduces well the trends and the averaged magnitude of wind speed in Rotsee. It allows a better estimation of wind speed than the Lucerne measurements by decreasing the root mean square error from $E_{RMS} = 1.6$ m s$^{-1}$ to $E_{RMS} = 0.67$ m s$^{-1}$. The distribution of wind speed in Rotsee is also better reproduced with the NN estimates than with the Lucerne data (Fig. R1.2).

The effects of wind speed on the sensible and latent heat fluxes are taken into account in the calibration function $f$ (McJannet et al., 2012; Fink et al., 2014). Several empirical expressions for $f$ are available in the literature. We used $f = (2.33 + 1.65 U_w) L_{fetch}^{-0.1} + 0.26(T_w - T_a)$ based on McJannet et al. (2012) and Fink et al. (2014), with $U_w$ the wind speed at 2 m height, $L_{fetch} = 2500$ m the lake fetch, $T_w$ the lake surface temperature and $T_a$ the air temperature. This expression for $f$ was selected by comparing the estimated heat fluxes with the observed change of heat content at MB. An error of $\delta_w = 0.67$ m s$^{-1}$ in the wind speed leads to an error of $\delta_f = 1.4$ W m$^{-2}$ mbar$^{-1}$ in the function $f$, which is lower than the uncertainty of $f$ (differences between estimates of $f$ can reach 3 W m$^{-2}$ mbar$^{-1}$ depending on the empirical formula used). The resulting errors in the surface heat flux $H_{Q_0}$ and surface buoyancy flux $B_0$ depend on the meteorological forcing. From the yearly averaged meteorological data, the errors are $\delta_{H_{Q0}} = 6.1$ W m$^{-2}$ and $\delta_{B_0} = 2.1 \times 10^{-9}$ W kg$^{-1}$, which is 5 % of the yearly averaged $H_{Q_0}$ and $B_0$. We will include the error on the heat fluxes on line 152: "*The uncertainty in the estimates of wind speed and relative humidity leads to an average uncertainty of 5 % and 3 % of the surface heat fluxes (Sect. 2.2), respectively.*"

[Figure]

Figure R1.1: One month-long time series of wind speed measured by the Lucerne (LUZ) and Rotsee (ROT) stations, and estimated from Neural Network Fitting (NNF) from November to December 2020.

[Figure]

Figure R1.2: Box plots of the wind speed measured at the Lucerne (LUZ) and Rotsee (ROT) stations, and estimated from Neural Network Fitting (NNF). The box plots are based on the dataset of Fig. R1.1.

**References:**

Fink, G., Schmid, M., Wahl, B., Wolf, T., and Wüest, A.: Heat flux modifications related to climate-induced warming of large European lakes, Water Resour. Res., 50, 2072–2085, https://doi.org/10.1002/2013WR014448, 2014.

McJannet, D. L., Webster, I. T., and Cook, F. J.: An area-dependent wind function for estimating open water evaporation using land-based meteorological data, Environ. Modell. Software, 31, 76–83, https://doi.org/10.1016/j.envsoft.2011.11.017, 2012.

Philippopoulos, K. and Deligiorgi, D.: Application of artificial neural networks for the spatial estimation of wind speed in a coastal region with complex topography, Renewable Energy, 38, 75–82, https://doi.org/10.1016/j.renene.2011.07.007, 2012.

**RC3:** *Methods 2.2.*

*L 160-161. Is the SW measured or parametrized? From section 2.1 it seems that it is measures but then in 2.2 it seems to be parametrized. Being a widely available parameter, I do not see the need to compute it.*

**AR3:** Incoming solar radiation reaching the lake's surface ($R$) is directly measured (Sect. 2.1, line 140). However, a part of $R$ is reflected at the lake's surface and is not included in the shortwave radiation entering the lake $H_{SW,0}$. To compute $H_{SW,0}$, the albedo of direct and diffuse solar radiation is parametrized as a function of the cloudiness (Fink et al., 2014).

**Reference:**

Fink, G., Schmid, M., Wahl, B., Wolf, T., and Wüest, A.: Heat flux modifications related to climate-induced warming of large European lakes, Water Resour. Res., 50, 2072–2085, https://doi.org/10.1002/2013WR014448, 2014.

**RC4:** *L172. Are there measurements to support the S value? Did you perform any sensitivity to assure that possible variations had no effect on your evaluations at a seasonal scale?*

**AR4:** Yes, we have salinity estimates from conductivity profiles collected over the year. The surface salinity increases from summer to winter by approximately $\Delta S \approx 0.5$ g kg$^{-1}$ due to vertical mixing between the epilimnion and hypolimnion. The associated change of density is around $\Delta_S \rho \approx 0.4$ kg m$^{-3}$, which is almost one order of magnitude lower than the seasonal change of density due to surface temperature $\Delta_S \rho \approx 3$ kg m$^{-3}$. We suggest adding the following sentence on line 172: "*We*

*assume a constant salinity over the year, as the seasonal change of surface water density due to salinity is one order of magnitude lower than the seasonal change due to temperature.*"

**RC5:** *L176. From this paragraph it seems that HQ0 is always a negative loose term, while the shortwave is the only one term that contributes to heating. On the contrary LWin and HC can be positive too. In general the way to manage the signs of the fluxes in this section terms is a bit confusing. I suggest to reason in term of H0net and B0net only (1+2 eq), without distinguishing between SW and other terms. What is important to verify is whether the net flux is positive or negative. This suggestion should be extended to the other sections of the paper.*

**AR5:** All the heat fluxes are defined positive in the upward direction (cooling), as explained on lines 154-155. Even if some of the surface heat fluxes can be negative (heating) as mentioned by the reviewer, the total surface heat flux $H_{Q_0}$ remains indeed positive for most days. This continuous surface cooling is mainly due to the loss of longwave radiation. We believe that the confusion comes from lines 175-176 where we oppose surface cooling to radiative heating. We propose to modify this sentence as follows: *"The net buoyancy flux at the surface is $B_{0,net} = B_0 + B_{SW,0}$. $B_{0,net} > 0$ indicates a destabilizing buoyancy flux (net cooling) whereas $B_{0,net} < 0$ indicates a stabilizing buoyancy flux (net heating)."*

We agree with the reviewer regarding the use of the net heat and buoyancy fluxes. The cooling and heating phases must be determined from $H_{0,net}$, which is directly related to $B_{0,net}$ (lines 176-178). However, the driving force of TS is surface cooling, expressed by a destabilising surface buoyancy flux $B_0$. Distinguishing $B_0$ from the radiative (penetrative) buoyancy flux is required to determine the convective velocity scale (Fig. 4) and the transport scaling formulae (Sect. 2.6). The use of $B_{0,net}$ and $B_0$ is already explained on lines 176-178.

**RC6:** *Methods 2.4. In the methods aimed at detaching the thermal syphons I would have expected to see the vertical component of the velocity as a target variable, in particular before the beginning of the event. Is there any reasons why you did not mention it? Given the uncertainty on the wind data, I think it could be a better way to distinguish between thermal syphons and wind driven flow (see e.g. Fer et al. 2002).*

**AR6:** Thank you for the suggestion. The vertical velocity measured by the ADCP has indeed a different signature between TS events and wind-driven flows and we initially tried to use it in the "wind filter" of our algorithm. TS events are characterized by convective plumes with an alternating upward and downward vertical velocities (Fig. 3c), whereas wind-driven cross-shore flows are associated with strong downwelling at MT, when they are directed in the x-direction ($U_x > 0$). The challenging aspect of this approach is the definition of the criterion used to distinguish the two different signatures in the

vertical velocity. An option is to use a threshold value for $U_z$ (which is larger for downwelling than convection) or to focus on the change of sign of $U_z$ over a certain period. Regarding reproducibility, these criteria involve arbitrary threshold values that are not necessarily physically grounded and could be more system-dependent than our criterion based on the Monin-Obukhov length $L_{MO}$. We also think that vertical velocities can be more difficult to measure in the field than $L_{MO}$, as they require high resolution ADCP data. Additionally, the wind filter based on $L_{MO}$ can be applied a priori to predict the occurrence of TS on a specific day. Although there is uncertainty in the wind speed estimation, our wind filter correctly discarded all the cross-shore flows associated with downwelling.

We are not sure what the reviewer refers to in Fer et al. (2002). To our understanding, Fer et al. (2002) did not use the vertical velocity to distinguish between TS and wind-driven flows. They used the ratio $z/L_{MO}$ to study the effects of wind on TS (with $z$ the depth of interest, see their figure 7), which is similar to our approach. Yet, we still think that using vertical velocities could be an interesting approach to try in the future and we will mention it on line 641 in Appendix C: "*Additional filters could be implemented to distinguish between TS and wind-driven cross-shore flows, based for example on high-resolution vertical velocity measurements, observed oscillations of the thermocline (e.g., wavelet analysis), estimates of the period of internal waves and identification of upwelling events (e.g., Wedderburn and Lake numbers) (Imberger and Patterson, 1989).*"

**Reference:**

Fer, I., Lemmin, U. and Thorpe, S. A.: Winter cascading of cold water in Lake Geneva, J. Geophys. Res., 107(C6), 3060, https://doi.org/10.1029/2001JC000828, 2002.

**RC7:** *Table1. Separate the extremes of the range with a "-" in place of a ",". Why no range for Ux and Uz?*

**AR7:** We used "," to avoid confusion with the minus sign. But we will follow the reviewer's suggestion and use "-" for the ranges in Table 1, with brackets around negative values. We will also add the range of values for $U_x$ and $U_z$: $[(-0.05) - 0.07]$ m s$^{-1}$ and $[(-0.01) - 0.01]$ m s$^{-1}$, respectively.

**RC8:** *Figure 3,4,7,8. The figures of this paper are too much dense of information. The effort to make them fully informative has the counter-effect to confuse the reader with too much data and is not efficient in highlighting a clear message. Make an effort to make the figures clearer, with less but more direct information, and eventually reduce the number of figures. ( Fig. 8 in particular is really hard to follow)*

**AR8:** We appreciate the critical assessment of the figures provided by the reviewer. We acknowledge that our figures provide a lot of information but we tried to provide a detailed description in each caption. The two other reviewers are both very positive about the figures (R2: "*I (...) found all of the*

*figures engaging*", R3: "*The writing and presentation are mostly very good.*"). We will still try to improve the clarity of Fig. 3, 4, 7 and 8 as follows:

- Figure 3: see the specific answer AR9 below.
- Figure 4: we suggest removing the grey error bars in Fig. 4a to improve the readability.
- Figure 7: we will better indicate the month for each row of subpanels.
- Figure 8: we propose to move Fig. 8b to Appendix B as it adds complexity and it is not directly related to Fig. 8a. We will replace it with a schematic illustrating the different time scales over the cooling phase (see Fig. R2.2 in the response to Referee #2).

**RC9:** *Fig. 3. The first panel is useless. In the second panel limit the plot of B0net. Velocity and temperature contours together limit the readability. Look at Fig. 3 of Fer et al. 2002 as an example of a good representation of a single event: a single line of Ux and Uz is much clearer. Finally the contours of temperature between 13.00 and 17:00 looks like affected by an error in the interpolation. Do you have enough thermistors? If so, how do you explain the different pattern? How do you explain the rapid changes in signs of Uz? Please comment.*

**AR9:**

- Fig. 3a: We do not think that this panel is useless, as it illustrates the seasonality of the diurnal cycle shown for a specific day in the three other panels. In particular, Fig. 3a shows the seasonality of (1) the duration of the heating and cooling phases and (2) the magnitude of the forcing. The seasonality of the diurnal cycle is a key aspect to understand how the occurrence (Sect. 4.2) and the flushing period (Sect. 4.4) vary over the year. We will add more references to this panel in the text. We also think that the reviewer's comment comes from the fact that Fig. 3a seems disconnected from the three other panels. We will better link them by clearly indicating in Fig. 3a where the diurnal cycle of Fig. 3b-d appears.

- Fig. 3b: Showing only $B_{0,net}$ does not provide enough information in our view. $B_{0,net}$ implicitly indicates the cooling and heating phases, associated with destabilising and stabilising surface buoyancy fluxes, respectively. However, $B_0$ and $B_{SW,0}$ are relevant to demonstrate that the main driver of the diurnal cycle is the solar radiation, and not the temporal change of surface cooling ($B_0$ remains positive all day). To better emphasize $B_{0,net}$, we will decrease the linewidth of $B_0$ and $B_{SW,0}$.

- Fig. 3c-d: The goal of this figure is not only to identify a TS event as in Fig. 3e of Fer et al. (2002b), but also to provide the main characteristics of the convective circulation (opposite cross-shore flows, thickness of TS, region of maximum velocity) and to indicate the region where the transport is calculated (blue curves in Fig. 3d). The latter requires to show velocity contours, and not only depth-averaged velocities. This is similar to the event presented in Fig.

2 of Fer et al. (2002a). Note that Fig. 3e of Fer et al. (2002b) focuses on the cross-shore velocity $U_x$ and the along-shore velocity $U_y$. It does not include the vertical velocity $U_z$ as in our Fig. 3c.

We also want to keep the isotherms in Fig. 3d, as they show the stratification induced by the density current. To improve the readability, we suggest increasing the spacing between the isotherms and showing them in gray. The dense surface temperature contours during the second day are due to the strong surface heating captured by the surface thermistor. The temperature has been linearly interpolated between the thermistors and there is no thermistor between the surface and 3 m depth due to rowing restrictions (lines 124-126 and Table A1), which leads to dense isotherms down to 3 m depth (2 m height above the bottom). We will add a tick on the y-axis to indicate the location of the surface thermistor and we will mention the linear interpolation of temperature in the caption.

Vertical velocities in Fig. 3c have a temporal resolution of 15 min, as explained in the figure caption. We did not interpolate them over time: one value is shown at each depth every 15 min. The "rapid" changes in signs of $U_z$ come from the upward-downward motion of the convective plumes every 15 min. The duration of half of a convective overturn can be estimated as $\tau_{conv} = h_{MT}/U_z$ with $h_{MT} \approx 4$ m the depth at MT and $U_z \approx 0.005$ m s$^{-1}$ the velocity of convective plumes. This gives $\tau_{conv} \approx 13$ min, which is less than the temporal resolution of Fig. 3c. The convective plumes can thus have opposite directions between two consecutive measurements.

**References:**

Fer, I., Lemmin, U. and Thorpe, S. A.: Contribution of entrainment and vertical plumes to the winter cascading of cold shelf waters in a deep lake, Limnol. Oceanogr., 47(2), 576–580, https://doi.org/10.4319/lo.2002.47.2.0576, 2002a.

Fer, I., Lemmin, U. and Thorpe, S. A.: Winter cascading of cold water in Lake Geneva, J. Geophys. Res., 107(C6), 3060, https://doi.org/10.1029/2001JC000828, 2002b.

**RC10:** *Results 3.4. The R2 values are really low, I would limit the analysis only to the variables which show at least a trend (not the case of tauf for example). In the conclusion you defined "robust" these relationships , but these R2 do not support these conclusions. I would be more cautious to base the conclusions on the basis of these results.*

**AR10:** We acknowledge that the $R^2$ values are low, which is not surprising knowing the natural variability of the process. We attributed the strong variability of $q_{avg}$ and $\tau_F$ to the fluctuations of the thickness of TS, as discussed in Appendix C (lines 655-663). Despite the scatter between days, the four

variables of Fig. 6, including $\tau_F$, show a linear trend. This is confirmed by the low p-value ($p_{val,F} <$ $10^{-53}$ for the four variables), which indicates that the slope of the linear fits is significantly different from zero. We agree with the reviewer that the term "robust" may be excessive: we will remove it from the conclusion. Lines 588-589 will now read: "*This study provides a field validation of laboratory and theoretically based scaling.*"

**RC11:** *References. A careful review of the references is needed (for example Rao and Schwab, Meyers and Dale are not present)*

**AR11:** Thank you for the comment, we apologize for this issue. We realized that our reference management software did not work properly. We will add the following missing references:

- Fink, G., Schmid, M., Wahl, B., Wolf, T., and Wüest, A.: Heat flux modifications related to climate-induced warming of large European lakes, Water Resour. Res., 50, 2072–2085, https://doi.org/10.1002/2013WR014448, 2014.
- McJannet, D. L., Webster, I. T., and Cook, F. J.: An area-dependent wind function for estimating open water evaporation using land-based meteorological data, Environ. Modell. Software, 31, 76–83, https://doi.org/10.1016/j.envsoft.2011.11.017, 2012.
- Meyers, T. and Dale, R.: Predicting daily insolation with hourly cloud height and coverage, J. Climate Appl. Meteor., 22, 537–545, https://doi.org/10.1175/1520-0450(1983)022<0537:PDIWHC>2.0.CO;2, 1983.
- Rao, Y. R. and Schwab, D. J.: Transport and mixing between the coastal and offshore waters in the Great Lakes: a review, J. Great Lakes Res., 33, 202–218, https://doi.org/10.3394/0380-1330(2007)33[202:TAMBTC]2.0.CO;2, 2007.

---

## Author Comment (AC2)

**Interactive discussion on "Seasonality of density currents induced by differential cooling"**

(Tomy Doda, Cintia L. Ramón, Hugo N. Ulloa, Alfred Wüest, and Damien Bouffard)

**Response to Referee #2**

**R (Referee):**
*HESS three Prinicipal Review Criteria*

*1. Scientific significance is excellent with new concepts, methods, and data.*

*2. Scientific quality is good. The approach and applied methods are valid and the results are appropriately discussed.*

*3. Presentations quality is good. There is an appropriate number and quality of figures/tables, appropriate use of English language. I found that there were few typos and that the English is good. The use of some symbols and abbreviations is confusing.*

*General Comments*

*The preprint addresses an important aspect of differential cooling with new concepts and data. With an extensive data set the thermal siphon process is shown to flush the near shore region. A simple model based on practically measured or available data is used to predict this process and it's seasonal variability. I enjoyed reading all five sections and the appendices and found all of the figures engaging. In summary I beleive the work represents a significant contribution to the field and is well suited to HESS.*

**A (Authors):** We are grateful to Referee #2 for his/her positive feedback and for his/her comments about the clarity of the manuscript. We address them below.

*My only general comment is that the clarity of the paper's main findings are obscured somewhat by the complex collection of abbreviations and symbols. In the first three specific comments below I address this and other clarity issues that I think should be addressed.*

*Specific comments*

*Three specific comments related to the overall clarity:*

**RC1 (Referee's Comment #1):**
*The along-x locations and their labels are confusing even after the reader is comfortable with the XZ description of the lake:*

*$L\_lit$ is the distance from one end of the lake along the thalweg to MT. So MT was located at the location where the photic zone reaches the bottom? I don't think this is ever stated, rather it seems MT is located at an arbitrarily shallow location along the thalweg.*

*$L\_ML$ the distance from the same origin along the thalweg to the isobath that matches the depth of the mixed at MB.*

*$x\_q$ the distance from the same origin to MT (where q is measured/predicted but not clearly linked between Figures 1 and 2).*

*$l\_p$ the length of the plateau (not indicated in Figure 2).*

*I think the formatting of these labels should be more consistent (e.g. a capital letter L followed by a subscript) and that $x\_q$ or $L\_lit$ be omitted. A similar simplification would help with the depths ($d\_p, d\_MT, h\_TS, h\_lit$, etc). I never could figure out what MT and MB stood for.*

**AR1 (Authors Response #1):** Regarding the notations, we will follow the reviewer's suggestion and use a capital letter $L$ for distances along $x$ ($L_{lit}$, $L_{ML}$, $L_p$) and a lower case $h$ for distances along $z$ ($h_p$, $h_{MT}$, $h_{TS}$, $h_{lit}$, $h_{ML}$). The letter B in MB stands for "Background". The letter T in MT stands for "Thermal siphon". We will add these definitions to lines 119-120: "*We monitored the background stratification at the deepest location ("background mooring" MB, approx. 16 m deep) as well as the dynamics of TS offshore from the plateau region ("TS mooring" MT, approx. 4 m deep), from March 2019 to March 2020 (Fig. 2a).*"

Regarding the specific length scales:

- Indeed, $L_{lit}$ is the length of the littoral region flushed by TS at MT. As discussed in Sect. 4.3 (lines 521-526), this length scale depends on the location of measurements along $x$ because it is used to parametrize the $x$-dependent flushing time scale.

  The location of MT is constant over the entire year and we never stated in the manuscript that it was selected on a photic-depth criterion (which is seasonally dependent). The photic zone, estimated from repeated Secchi Depth measurements, was deeper than the water depth at MT except during the productive period in late summer 2019, where the photic depth was reduced to ~ 4 m.

  MT was positioned along the thalweg, with the two following criteria: (1) to be in the sloping region and (2) to be shallower than the mixed layer depth during cold summer nights ($h_{ML} \approx 5$ m). These two conditions allowed us to capture downslope TS already in summer.

  We propose to add the following sentence on line 120: "*The mooring MT was located along the thalweg, at the beginning of the sloping region. This shallow water column is already vertically mixed in summer by the action of surface cooling.*"

- $L_{ML}$ is defined from the mixed layer depth because it corresponds to the distance over which differential cooling takes place (Sect. 4.3, lines 518-521).

- We used $x_q$ to refer to the location of discharge measurements in other studies (Table 2). In our case, the length of the littoral region $L_{lit}$ is equal to $x_q$. We agree that these different notations might be confusing for the reader and we will replace $x_q$ by $L_{lit}$ in Table 2. We will specify in the caption that $L_{lit}$ is the length of the flushed littoral region, defined based on the location of discharge measurements.

- We will add the $x$-axis, $L_p$ and $L_{lit}$ in Figure 2.

**RC2:** *Although the transect data in the schematic represents an efficient use of space and looks great I think it unnecessarily complicates the schematic. The schematic should address the seasonal cycle, identify the plateau, perhaps include the equation q=c_q h (BL)^1/3 or similar equation for U, and serve as a road map or foreshadowing for the rest of the paper. Something like Table 2 added to the introduction could compliment the schematic. Where is the origin x=0 on the map in Figure 1? Why not identify the plateau in the schematic? Could the authors incorporate a graphic illustrating the essential time scales? If aspects at the end of the paper are too complicated to include in the initial schematic provide a revised schematic at the end of the results or in the discussion. I recognize the authors have spent some time linking the text and figures including Figure 1 and Table 1 together but it still needs improvement.*

**AR2:** Thank you for helping us improve Fig.1, we propose a revised version of the figure below (Fig. R2.1).

We would like to keep the transect data in Fig. 1, as it provides important information on (1) differential cooling and (2) TS-induced stratification. However, we understand the concern of the reviewer about the clarity of the figure. To simplify the schematic, we suggest keeping the colormap but removing the isotherms and replacing the vertical dashed lines with points on the $x$-axis.

As suggested by the reviewer, we will add the plateau to the schematic and indicate $L_p$ and $h_p$. We propose to use four boxes on top of the schematic to list the scales related to the littoral region, the plateau region and the thermal siphon, and to address the seasonal cycle with a conceptual graphic showing the seasonality of the forcing conditions ($B_0$, $h_{ML}$, $L_{ML}$). We will also include the scaling formulae for $U$, $q$ and $\tau_F$ (Eqs. (9), (10), (11)) in the box about TS.

The origin $x = 0$ is already indicated in Fig.1. We believe that the reviewer is referring to the bathymetric map of Fig. 2. We will add the x-axis and its origin in Fig. 2 (see AR1).

The time scales are currently introduced in different sections: Sect. 3.2 ($\tau_c$), Sect. 3.5 ($\tau_t$) and Sect. 4.2 ($\tau_{ini}$, $\tau_{mix}$). We realized that this can be confusing for the reader and it might be the reason why the reviewer is asking for an overview of the time scales in Fig. 1. We propose to introduce $\tau_c$, $\tau_{ini}$ and $\tau_t$ in Sect. 2.6, as they are based on previous studies. We will keep $\tau_{mix}$ in Sect. 4.2 as it is a modification of $\tau_{ini}$ that we propose in this study. It is difficult to include these time scales in Fig. 1 since the current schematic shows the spatial and not the temporal variability of TS. We will mention the initiation time scale in the box about TS but we prefer to illustrate the other time scales in Fig. 8, once they all have been introduced in the text. We will move Fig. 8b to the Appendix and replace it with a schematic illustrating the time scales $\tau_{ini}$, $\tau_{mix}$, $\tau_t$ and $\tau_c$ over the cooling phase. The revised version of Fig. 8 is shown below as Fig. R2.2.

We hope that the different changes mentioned above will help to better link the text to the figures.

[Figure]

Figure R2.1 (revised Fig. 1): Data-based schematic of the cooling-driven thermal siphon representing the plateau, littoral and mixed regions, the seasonality of the forcing and the variables used for the transport scaling. The littoral region is the region upslope of MT, where the current velocity is measured and transport variables are calculated. The cross-shore temperature field is linearly interpolated from a transect of CTD (Conductivity-Temperature-Depth) profiles collected in the morning on 22 August 2019 (08:20–08:50 UTC), from x = 225 m to x = 714 m. Black dots on the x-axis show the location of the profiles. The green dashed line in the seasonality diagram corresponds to the transition period between the mixing period (winter) and the stratified period (summer), when there is not a well-defined mixed layer.

[Figure]

Figure R2.2 (revised Fig. 8): Time scales determining the occurrence of TS. (a) Schematic of the three periods of the cooling phase at MT parametrized by the mixing time scale $\tau_{mix}$, transition time scale $\tau_t$ and cooling duration $\tau_C$. The mixed layer depth is expressed as the relative depth $h_{ML}{}' = h_{ML} - h_{MT}$, with respect to $h_{MT} = 4$ m. The mixed layer deepens during the first period ($0 < t < \tau_{mix}$), until the complete mixing of the water column. Convection dominates over the second period ($\tau_{mix} < t < \tau_{ini}$). TS occurs during the third period ($\tau_{ini} < t < \tau_c$). (b) Effects of the seasonality of $\tau_{mix}, \tau_t, \tau_C$ and $h_{ML}{}'$ on the occurrence of TS. Monthly averages are represented, with shaded areas ($\tau_C, \tau_t, \tau_{mix}$) and error bars ($h_{ML}{}'$) indicating the monthly standard deviation. Note the log-scale for the axis of timescales. The gray shaded period corresponds to optimal conditions for the occurrence of TS.

**RC3:**

*The language related to flow direction is sometimes confusing. I think this is partially due to the fact that the shorelines to the northwest and southeast are closer to both stations than the shore line to the northeast. The authors should explicitly state early in the paper and repeat in several captions that offshore flow is southwestward flow or something similar, line 191 is inadequate. I don't think the authors ever comment on along shore flow, tell the reader why it's ignored or if there's none.*

**AR3:** Thank you for this comment, we indeed need to better introduce the framework that we are using regarding the flow direction. Our motivation in this study is to quantify the flushing of the littoral plateau region at the north-eastern end of the lake. We use a 2D framework to study the convective circulation (Fig. 1) and focus on the TS-induced transport along the thalweg (x-axis). The x-axis defines what we call the "offshore direction". Due to its elongated shape, Rotsee is suitable for this 2D framework, with the strongest TS flowing preferentially along the thalweg. The along-shore flow (y-axis) is generally small but not necessarily zero as TS can be slightly deviated from the x-axis by Coriolis or topographic effects. The 3D aspects of TS are out of the scope of this study but could be investigated with 3D numerical experiments.

We suggest explaining our 2D framework and clarifying the language about the flow direction in Sect. 2.1, by adding the following paragraph after line 113: *"In this study, we focus on quantifying TS originating from the north-eastern plateau region (Fig. 2a). Because of the elongated shape of Rotsee, we use the 2D (x, z) framework shown in Fig. 1 by orienting the x-axis along the thalweg. We assume that TS originating from the plateau region preferentially flows along the x-axis and we do not consider flows in the perpendicular direction. We will now refer to the north-eastern end of the lake as the "shore" and call the direction of the x-axis the "offshore direction"."*

In addition, we will also modify lines 190-192 to define the cross-shore velocity: *"The horizontal velocity was projected onto the x-axis (angle of 56° from north), which crosses the isobath at MT perpendicularly (Fig. 2). Following the 2D framework of Fig. 1, we will now call the velocity $U_x$ the "cross-shore velocity"."* We will specify "southwestward flows" where we refer to cross-shore flows in the captions of Figs. 3 and 5.

**RC4:** *I was expecting to see more transects demonstrating the TS during other times of the year e.g. TS in July, October and December, were there no others collected?*

**AR4:** Transects of temperature profiles were collected during twelve campaigns from August to December 2019. We used one transect in Fig. 1 to show the cross-shore temperature distribution. We did not include the other transects in the manuscript because they do not clearly show the seasonality of TS. The bottom stratification and the TS thickness are similar between transects. The main seasonal differences are the depth and length of the mixed littoral region ($L_{ML}, h_{ML}$) and the duration of the TS events. Temperature transects are more relevant to study the short-term variability of TS over one diurnal cycle (periods shown in Fig. 3d), which is not the objective of this study.

We believe that the reviewer expected to see more transects because we mentioned the different campaigns on lines 133-135. We will remove this unnecessary information and modify the sentence as: "*To capture the spatial variability of TS, cross-shore transects of Conductivity-Temperature-Depth (CTD) profiles (Sea&Sun CTD 60M, sampling interval of 0.4 s) were performed along the x-axis.*"

**RC5:** *Figure 2. Provide the depth at MT in Figure 2 (b). I think the map of Switzerland should be idienfied as a map of Switzerland.*

**AR5:** The schematic of the mooring in Fig. 2b refers to both MB and MT. We propose to indicate the depth of both moorings on the map of Fig. 2a. We will mention the map of Switzerland in the caption as follows: "*The location of Rotsee is shown on the map of Switzerland with a black dot.*"

**RC6:** *Lines 145 to 152 - has anyone ever done this before for winds or humidity? explain why you think the simpler approach failed. Can you provide a separate R^2 for the northerly and westerly wind components, or the along and across axis wind components?*

**AR6:** Artificial Neural Networks are commonly used for the spatial interpolation of meteorological parameters, including wind speed (Öztopal, 2006; Kusiak and Li, 2010; Philippopoulos and Deligiorgi, 2012) and relative humidity (Yasar et al., 2012; Philippopoulos et al., 2015). Unlike pressure, air temperature and solar radiation, a simple linear interpolation cannot be used for wind speed and relative humidity because these two parameters are highly variable over time and are dependent on the surrounding environment. Philippopoulos and Deligiorgi (2012) showed for instance that Neural Networks are more performant than traditional interpolation methods of wind speed.

The values of $R^2$ and $E_{RMS}$ for cross-shore (x-axis) and along-shore (y-axis) wind components are $R_x{}^2 \approx 0.85$, $E_{RMS,x} \approx 0.61$ m s$^{-1}$ and $R_y{}^2 \approx 0.64$, $E_{RMS,y} \approx 0.26$ m s$^{-1}$, respectively. We will not include this information in the manuscript, as we did not use the wind direction in the analysis.

**References:**

Kusiak, A. and Li, W.: Estimation of wind speed: A data-driven approach, J. Wind Eng. Ind. Aerodyn., 98, 559–567, https://doi.org/10.1016/j.jweia.2010.04.010, 2010.

Öztopal, A.: Artificial neural network approach to spatial estimation of wind velocity data, Energy Convers. Manage., 47, 395–406, https://doi.org/10.1016/j.enconman.2005.05.009, 2006.

Philippopoulos, K. and Deligiorgi, D.: Application of artificial neural networks for the spatial estimation of wind speed in a coastal region with complex topography, Renewable Energy, 38, 75–82, https://doi.org/10.1016/j.renene.2011.07.007, 2012.

Philippopoulos, K., Deligiorgi, D., and Kouroupetroglou, G.: Artificial Neural Network modeling of relative humidity and air temperature spatial and temporal distributions over complex terrains, in: Pattern Recognition Applications and Methods, vol. 318, edited by: Fred, A. and De Marsico, M., Springer International Publishing, Cham, 171–187, https://doi.org/10.1007/978-3-319-12610-4_11, 2015.

Yasar, A., Simsek, E., Bilgili, M., Yucel, A., and Ilhan, I.: Estimation of relative humidity based on artificial neural network approach in the Aegean Region of Turkey, Meteorol. Atmos. Phys., 115, 81–87, https://doi.org/10.1007/s00703-011-0168-2, 2012.

**RC7:** *Line 239 I think this is ok for B_0 and is discussed later but I'm not so sure about L_ML, wouldn't this often increase over the cooling period?*

**AR7:** Yes, the mixed layer can deepen by ~3 meters during intense daily cooling periods in late summer, which leads to an increase of $L_{ML}$ of ~70 meters over the same periods. This increase is more limited in late autumn, due to the weaker convection. We are averaging $L_{ML}$ over the cooling phase since we are interested in the estimation of daily averaged transport variables only. We are not investigating here the short-term temporal changes of $U$ and $q$ over the cooling phase. Moreover, the daily increase of $L_{ML}$ changes the velocity scale $(B_0 L_{ML})^{1/3}$ by O(10$^{-3}$) m s$^{-1}$, which is one order of magnitude lower than $(B_0 L_{ML})^{1/3}$. For typical summer conditions with $B_0$~10$^{-7}$ W kg$^{-1}$ and $L_{ML}$~200 m for example, an increase of $L_{ML}$ by 70 meters changes the velocity scale $(B_0 L_{ML})^{1/3}$ by ~0.003 m s$^{-1}$.

**RC8:** *Table 1 would benefit from some recomposition to aid in connecting the four columns, particularly the fourth column, e.g. swap the third and fourth column and justify the 'definition and equation' column left.*

**AR8:** We will follow the reviewer's suggestion. The ranges of values will be provided in the third column and the equations will be in the fourth column and justified to the left.

*Technical corrections*

**RC9:** *Whether limnology is patriarchal or not the reference to 'fathers of limnology' reads a little too patriarchal.*

**AR9:** We will replace "fathers of limnology" by "pioneer limnologists".

**RC10:** *The whole sentence beginning 'Such shift' on line 594 needs improvement, to start, change 'Such shift' to 'Such a shift'.*

**AR10:** We propose to modify this sentence as follows: "*Such a timing has implications for the transport of dissolved compounds, with, for instance, stronger exchange between littoral and pelagic waters at a time of high primary production (summer and daytime).*"

**RC11***: line 373 and 374 change shadow to shading.*

**AR11:** We will change the two occurrences of "shadow" to "shading".

**RC12:** *line 376 refer to figure 7 for the histograms.*

**AR12:** We will add the reference to Fig. 7.

**RC13:** *line 475 remind the reader what the depth is at MT.*

**AR13:** We will specify the depth of MT in the caption of Fig. 8 (see the caption of Fig. R2.2).

---

## Author Comment (AC3)

**Interactive discussion on "Seasonality of density currents induced by differential cooling"**

(Tomy Doda, Cintia L. Ramón, Hugo N. Ulloa, Alfred Wüest, and Damien Bouffard)

**Response to Referee #3**

**R (Referee):**

*In this manuscript, the authors use a unique year long time series to estimate the frequency and strength of the thermal siphon in lakes and the influence on flushing of the littoral zone. This is an interesting dataset and addresses an important concept in physical limnology. The writing and presentation are mostly very good.*

**A (Authors):** We thank Referee #3 for his/her encouragement and his/her interesting questions about the interpretation of the results. We address all the comments from Referee #3 below.

*I have a few general comments that I would like to see addressed though:*

**RC1 (Referee's Comment #1):** *What is the magnitude of the outflow near the study site? Is it important relative to magnitude of the flushing rates?*

**AR1 (Authors Response #1):** There is no in-situ measurement in the outflow but the discharge can be estimated from the simulations of the Swiss river network (dataset MQ-GWN-CH from the Federal Office for the Environment). The monthly averaged simulated discharges of the Rotsee outflow in $m^3$ $s^{-1}$ are available here:

https://api.geo.admin.ch/rest/services/ech/MapServer/ch.bafu.mittlere-abfluesse/67200/extendedHtmlPopup?lang=en.

The average discharge is $\sim 0.1$ $m^3$ $s^{-1}$, which corresponds to a specific-width discharge at MT (total width of 150 m) of $q_{out} \sim 7 \times 10^{-4} m^2$ $s^{-1}$. This estimate is more than one order of magnitude smaller than the TS discharge $q_{avg} \sim 10^{-2} m^2$ $s^{-1}$ (Fig. 6). The effects of the outflow on the TS dynamics are thus negligible. We propose to mention the low discharge of the inflow and outflow on lines 108-109 as follows: *"The main in- and outflows are located at the south-western and north-eastern ends of the lake, respectively, and have a low discharge of $\sim 0.1$ $m^3$ $s^{-1}$."*

**RC2:** *What influence does the three-dimensionality of the littoral zone play? From my understanding, the entire framework here is 2D, but how uniform do you suppose q is across the lake? What are the limits of your results for other lakes in that context? It appears that Rotsee is about 2D as it gets, but is there a littoral zone aspect ratio where this all falls apart?*

**AR2:** The reviewer is correct: we used a 2D framework in this study and we focused on the lateral transport along the x-axis, as our objective was to quantify the flushing of the littoral plateau region at the north-eastern end of the lake. The elongated shape of Rotsee is suitable for this 2D framework, which is similar to the nearly 2D sidearm circulation observed in reservoirs (Adams and Wells, 1984; Monismith et al., 1990). We decided to take our measurements along the lake thalweg, which is the preferential direction of TS according to 3D numerical simulations (Ramón et al., 2019). We expect $q$ to be lower if it is measured away from the thalweg, closer to the north-eastern or south-western shores. The validity of such a 2D framework has to be further evaluated in more complex nearshore systems that might deviate from the conceptual model adopted here. Coriolis effect and local bathymetry perturbations might have to be included in these cases. For instance, the downslope flows observed by Fer et al. (2002) in Lake Geneva are not perpendicular to the shore due to spatial irregularities of the littoral region. The 3D aspects of TS are out of the scope of this study, but could be a motivation for future work. In particular, the effect of the littoral zone aspect ratio on the TS dynamics could be investigated with 3D numerical simulations.

We will add a paragraph after line 113 to explain our 2D framework. We refer to the answer AR3 to Referee #2 for more details about this paragraph. We also propose to add a few sentences about the 3D effects in other lakes after line 573: *"Finally, the 2D framework of TS requires specific validation in more complex nearshore systems and large lakes, where the topography, large-scale circulation and Coriolis may also affect the TS dynamics (Fer et al., 2002b). In these systems, the along-shore velocity component of TS must be considered in the cross-shore transport analysis."*

**References:**

Adams, E. E. and Wells, S. A.: Field measurements on side arms of Lake Anna, Va., J. Hydraul. Eng., 110, 773–793, https://doi.org/10.1061/(ASCE)0733-9429(1984)110:6(773), 1984.

Fer, I., Lemmin, U., and Thorpe, S. A.: Winter cascading of cold water in Lake Geneva, J. Geophys. Res., 107, 3060, https://doi.org/10.1029/2001JC000828, 2002.

Monismith, S. G., Imberger, J., and Morison, M. L.: Convective motions in the sidearm of a small reservoir, Limnol. Oceanogr., 35, 1676–1702, https://doi.org/10.4319/lo.1990.35.8.1676, 1990.

Ramón, C., Doda, T., Ulloa, H., and Bouffard, D.: Density currents induced by night-time cooling: offshore transport of littoral waters, in: Geophysical Research Abstracts, Vienna, Austria, 7-12 April 2019, EGU2019-970, 2019.

**RC3:** *Given the time of year where TS is most prevalent, does this play a role in accelerating autumn turnover?*

**AR3:** This is an interesting point. Yet, the effects on autumn turnover are out of the scope of our study and we will not discuss them in the manuscript. The intrusion of TS at the base of the mixed layer modifies the vertical thermal structure by bringing cold water above the thermocline. This advective heat flux enhances the cooling of the surface layer and should accelerate the deepening of the mixed layer in autumn, as observed for differential heating under ice (Ulloa et al., 2019). The shear induced by the intrusion of TS might also increase vertical mixing and lead to a faster erosion of the stratification (Strang and Fernando, 2001). However, the stratification induced by TS can prevent the water column to become entirely mixed in winter. A steady mixed layer can remain if the volume flux provided by TS balances the volume flux of mixed layer deepening (Wells and Sherman, 2001). As a result, the time of complete overturn would be delayed by TS and would only occur when TS stops later in winter (i.e., when $\tau_t > \tau_c$). In the case of Rotsee, TS rarely reaches the lake center and we do not expect a basin-scale effect on the autumn turnover. The effect should be more pronounced in lakes with a larger ratio $A_S/A_D$, where $A_S$ and $A_D$ are the surface areas of the shallow and deep regions, respectively (Wells and Sherman, 2001).

**References:**

Ulloa, H. N., Winters, K. B., Wüest, A., and Bouffard, D.: Differential heating drives downslope flows that accelerate mixed-layer warming in ice-covered waters, Geophys. Res. Lett., 46, 13872–13882, https://doi.org/10.1029/2019GL085258, 2019.

Strang, E. J. and Fernando, H. J. S.: Entrainment and mixing in stratified shear flows, J. Fluid Mech., 428, 349–386, https://doi.org/10.1017/S0022112000002706, 2001.

Wells, M. G. and Sherman, B.: Stratification produced by surface cooling in lakes with significant shallow regions, Limnol. Oceanogr., 46, 1747–1759, https://doi.org/10.4319/lo.2001.46.7.1747, 2001.

**RC4:** *TS is stronger in the summer, but less frequent. As noted, the summer is when this physical process might have the most impact on ecological and biogeochemical processes. Is there a way to compare TS across the months in a more quantitative fashion, for example, the total volume of water flushed in each month?*

**AR4:** We did not include the flushed volume in the manuscript because its estimation depends on the definition of the flushing period, which is arbitrary (Sect. 2.5, lines 211-212). It is still interesting to compare the seasonality of the daily flushed volume with the seasonality of the occurrence and intensity of TS. We estimated a daily average unit-width volume by normalizing the total volume of water flushed

every month by the number of measurement days (Fig. R3.1). The seasonal trend is similar to the occurrence (Fig. 5). This indicates that the weakening of TS from summer to autumn is overcome by the increase of occurrence and flushing duration. Overall, the occurrence of TS is the primary factor driving the seasonality of the flushing in Rotsee, whereas the seasonality of the discharge plays a secondary role. We propose to add Fig. R3.1 to the Appendix.

[Figure]

Figure R3.1: Daily averaged unit-width volume flushed by TS every month. Months with less than 10 days of measurements have been removed, as in Fig. 5 of the manuscript.

**RC5:** *Figure 3d - the contours on the upper right corner look more like an artefact of the contouring than anything that might possibly be real?*

**AR5:** These dense isotherms are due to the linear interpolation of the strong surface heating captured by the surface thermistor. We will better explain it in Fig. 3d, and we refer to the answer AR9 to Reviewer 1 for more details.

**RC6:** *Figure 6 - are you forcing the intercept here? From the equations that seems the case (an intercept of 0), but that doesn't look like the best fit line, at least for (b)*

**AR6:** Yes, we are forcing the intercept to be zero for the four quantities in Fig. 6. We agree that this approach does not provide the best fit but it follows the scaling formulae. The results of the linear fitting with non-zero intercept are:

$U_{avg} = 0.35 \cdot (B_0 L_{ML})^{1/3} - 8 \times 10^{-4}$ m s$^{-1}$,

$U_{max} = 1.22 \cdot (B_0 L_{ML})^{1/3} - 0.006$ m s$^{-1}$,

$q_{avg} = 0.28 \cdot h_{lit}(B_0 L_{ML})^{1/3} + 0.003$ m$^2$ s$^{-1}$ ,

and $\tau_F = 2.09 \cdot L_{lit}(B_0 L_{ML})^{-1/3} + 2.99$ h.

Note that those results are not significantly different from the scaling with a forced intercept. We will mention the zero intercept in the caption of Fig. 6. The lines 367-368 will now read: *"The equation of the linear regressions (with forced intercept to zero), the coefficient of determination ($R^2$) and the p-value of an F-test ($p_{val,F}$) are indicated."*

---

## Author Response (AR1)

**Author's response**

**"Seasonality of density currents induced by differential cooling"**

(Tomy Doda, Cintia L. Ramón, Hugo N. Ulloa, Alfred Wüest, and Damien Bouffard)

**Response to Editor**

**Editor:**
*I found the paper results very well presented though do agree with R1 that some polish on the editorial/expression aspects would just finalise this paper and ensure this good work is in the best light.*

*Whilst reading the discussion/conclusion, I wondered a) if our numerical models would typically resolve the nuance here (and maybe this is related to questions about errors in wind forcing?) and b) if a warming climate would potentially change the dynamics presented here. No doubt these are studies in their own right, but I thought this could also highlight the significance. For example, the end of Section 4.5 could comment on whether numerical models need further testing in this regard and could be used to help generalise across more complex bathymetries? The other "hot-topic" where we see papers is related to climate change effects to stratification phenology, and so worth highlighting this as something that may interact with TS? In fact, many 1D models are used to simulate climate change, but I often worry they may not capture shifts to dynamics like as described here. I wonder if it would be valuable to mention these issues.*

**A (Authors):** We thank the editor for his encouragement and his comments.

As further explained in the response to Reviewer #1, we have improved the readability of the manuscript by making the text more concise, re-organizing Sect. 4.2 and sending the manuscript for language editing.

Regarding question a), we are unsure whether the editor refers to 1D or 3D numerical models. One-dimensional numerical models are indeed not capable of resolving differential cooling since they do not take into account horizontal spatial variability. One of the objectives of the present study is to provide a parametrization of the TS-induced lateral transport that could be implemented in 1D models. As mentioned by the editor, this type of model is often used to study the effects of climate change and biogeochemical processes. Three-dimensional numerical models, however, reproduce well differential cooling and the TS-induced transport. They can be used for instance to investigate the effects of the slope on the TS formation (Ulloa et al., 2022) and the effects of external processes, such as wind, on lateral transport (Ramón et al., 2021). 3D numerical modelling is also relevant to study more complex

bathymetries. We have added a sentence about this aspect in lines 585-586: *"The effects of more complex bathymetries, departing from our 2D framework, could be further investigated with 3D numerical simulations."*

Regarding question b), we also think that it would be interesting to estimate the effects of climate change on the TS occurrence and intensity. This question could be addressed by performing numerical simulations with varying forcing conditions and stratification. It is difficult to predict the effects of climate change from our field-based study. We can expect that a longer and more intense stratified period due to climate change would decrease the occurrence of TS in summer/beginning of autumn by increasing the initiation time scale $\tau_{ini}$. Yet, for a given surface heat flux, higher surface temperatures increase the thermal expansivity $\alpha$, which leads to a higher buoyancy flux $B_0$ (Eq. (3)). An increase of $B_0$ could shorten the transition time scale (i.e., faster development of TS, Eq. (12)) and intensify the current (Eq. (10)). We have added a sentence about climate change in lines 593-594: *"The seasonality of TS may evolve in a changing climate, which also needs to be investigated. Changes in heat fluxes, summer stratification and surface temperature would affect both the intensity ($B_0$) and occurrence ($\tau_{ini}$) of TS."*

In the revised version of the manuscript, we have added the DOI number to our data repository. Note that the link will be activated once the review process will be finished (i.e., once all the figures will be definitive).

**References:**

Ulloa, H. N., Ramón, C. L., Doda, T., Wüest, A., and Bouffard, D.: Development of overturning circulation in sloping waterbodies due to surface cooling, J. Fluid Mech., 930, A18, https://doi.org/10.1017/jfm.2021.883, 2022.

Ramón, C. L., Ulloa, H. N., Doda, T., and Bouffard, D.: Flushing the lake littoral region: the interaction of differential cooling and mild winds, ESSOAR [preprint] https://doi.org/10.1002/essoar.10508544.1, 30 October 2021.

**Response to Referee #1**

**R (Referee):** *This paper describes an experimental work aimed at studying the occurrence of thermal siphons in Rotsee, a shallow lake sheltered by the wind. "Thermal syphon" indicates a physical process driven by differential cooling mainly due to bathymetry, which has important ecological implications, enhancing the hydraulic exchange between the littoral and the pelagic zone. The Authors made use of 1-year velocity and temperature data in the shallower area to detach thermal syphons. They focused on the frequency of occurrence over the year and analysed the forcing data suitable to explain this seasonality.*

*They developed a state-of-art experimental work, winding the situ descriptions of the phenomena, which are not so frequent in the literature, especially when aimed at investigating the process over the seasons.*

**A (Authors):** We thank Referee #1 for the critical assessment of the manuscript. His/her comments about the methods and the presentation of the results improved the clarity of the manuscript. We addressed them below.

**RC1 (Referee's Comment #1):** *The main limits of this contribution are the weak readability of the paper and the case-specific algorithm proposed to analyse the phenomena. With regard to the first aspect, my suggestion is to be much more concise in the text and in the figures, limiting the number of information to the most relevant ones, or alternative to help the reader to distinguish between the more and less relevant.*

**AR1 (Authors Response #1):**

- Regarding the readability of the paper, we followed the reviewer's suggestion and removed unnecessary information to make the text more concise. We moved the discussion about the diurnal variability of the occurrence of TS (Fig. 8b) to Appendix B (now Fig. B1). We also simplified Sect. 4.2 by introducing the initiation and transition time scales earlier in the methods (Sect. 2.6). We sent the manuscript for English language editing, which helped to gain clarity. Figs. 1, 2, 3, 4, 7 and 8 have been improved to provide only essential information, as explained in more detail in AR8 and AR9 and in the answer to Referee #2.

- Regarding the algorithm used to detect thermal siphon (TS) events, we acknowledge that it is lake-specific, as several criteria are based on threshold values that can vary between systems. Specificity of the algorithm has already been discussed in Sect. 4.5 and in Appendix D. However, such an algorithm can serve as a basis for detecting TS in other lakes. Hence, the study is not "case-specific" and the general structure of the algorithm (Sect. 2.4) can readily be

used in other systems. The only changes consist in adapting the threshold values and possibly modifying the filters to distinguish TS from other cross-shore flows. We modified Appendix D to better reflect this question. The lines 660-662 now read: *"The general structure of the algorithm can be used in other systems. Yet, several criteria are lake-specific and must be adapted to the system of interest. We discuss the limitations of the algorithm below and provide suggestions of improvement."*

We also added a few sentences in Sect. 4.5 (lines 571-574 in the revised manuscript): *"The general structure of the algorithm (Sect. 2.4) can serve as a basis for detecting TS in lakes, by adapting the lake-specific criteria to other systems. The 2D framework of TS requires specific validation in more complex nearshore systems and large lakes, where the topography, large-scale circulation and Coriolis may also affect the TS dynamics (Fer et al., 2002b). In these systems, the along-shore velocity component of TS must be considered in the cross-shore transport analysis."*

**RC2:** *More specific suggestions are listed in the followings.*

*Methods 2.1. The computation of the wind speed doesn't seem satisfactory for two reasons: 1- the location of the meteorological station is hardly well representative of the wind conditions over the lake's surface 2 – the methodology to derive wind data from the Lucerne station is not properly justified. Given the sheltering of this lake, a correlation between these sites' data is unlike and I am quite doubtful about the suitability of a neural network algorithm to estimate it, given the local character on the wind field. Despite I don't know the typical wind speeds at these site, I believe that in relative terms an error $E_{RMS}$ of 0.67 m/s is high. On the contrary, the approach is valuable for the other variables. I think that these sources of uncertainty must be accounted and discussed. Actually I do not have any suggestion regarding the solution of problem 1, apart for discussing this limitation in case of absence of other suitable data. With regard to problem 2, instead, I believe that is necessary to introduce an uncertainty in the fluxes evaluation.*

**AR2:**

- Regarding problem 1, we acknowledge that the wind speed can be different between the location of the meteorological station and the lake center (MB), although the distance between the two points is less than one kilometer. However, we think that our measurements are representative of the wind conditions over the nearshore plateau region, where thermal siphons are created (i.e., north-eastern end of the lake). The wind speed can vary spatially over the lake surface, even further offshore. The spatial variability of the wind speed is a common problem for the in-situ estimation of heat fluxes in lakes, as it cannot be resolved by a single meteorological station. In the case of Rotsee, we do not expect a significant effect of the spatial

variability of wind speed on the daily averaged heat and buoyancy fluxes, because of the small size of the lake and the surrounding topography channelizing the wind along the main axis of the lake. In addition, low wind conditions prevailed over the year. The daily-averaged wind speed was indeed less than 1 m s$^{-1}$ for 80 % of the days and less than 2 m s$^{-1}$ for 95 % of the days. We added a sentence about the assumption of spatial homogeneity in lines 169-170: "*We assumed that the meteorological conditions and the heat fluxes were spatially uniform over the lake surface (0.5 km$^2$).*"

- Regarding problem 2, we agree that estimating the wind velocity from the Luzern station introduces uncertainties, which we quantified by the root mean square error in lines 163-164 of the revised manuscript. However, we believe that a Neural Network (NN) approach is the most robust method to correct the Lucerne data to Rotsee. The performance of NN for estimating the spatial variability of wind speed has been demonstrated by Philippopoulos and Deligiorgi (2012). For other examples of studies using this approach, we refer to our answer AR6 to Referee #2.

To illustrate the performance of NN in Rotsee, we compared the estimated wind speed from NN with the measured wind speed in Lucerne and Rotsee stations over a month (Fig. R1.1). The period shown in Fig. R1.1 is not part of the NN training period. Although wind speed is larger in Lucerne than in Rotsee, a coherent correlation between the two sites is observed for most of the wind events. The NN approach reproduces well the trends and the averaged magnitude of wind speed in Rotsee. It allows a better estimation of wind speed than the Lucerne measurements by decreasing the root mean square error from $E_{RMS} = 1.6$ m s$^{-1}$ to $E_{RMS} = 0.67$ m s$^{-1}$. The distribution of wind speed in Rotsee is also better reproduced with the NN estimates than with the Lucerne data (Fig. R1.2).

The effects of wind speed on the sensible and latent heat fluxes are taken into account in the calibration function $f$ (McJannet et al., 2012; Fink et al., 2014). Several empirical expressions for $f$ are available in the literature. We used $f = (2.33 + 1.65U_w)L_{fetch}^{-0.1} + 0.26(T_w - T_a)$ based on McJannet et al. (2012) and Fink et al. (2014), with $U_w$ the wind speed at 2 m height, $L_{fetch} = 2500$ m the lake fetch, $T_w$ the lake surface temperature and $T_a$ the air temperature. This expression for $f$ was selected by comparing the estimated heat fluxes with the observed change of heat content at MB. An error of $\delta_w = 0.67$ m s$^{-1}$ in the wind speed leads to an error of $\delta_f = 1.4$ W m$^{-2}$ mbar$^{-1}$ in the function $f$, which is lower than the uncertainty of $f$ (differences between estimates of $f$ can reach 3 W m$^{-2}$ mbar$^{-1}$ depending on the empirical formula used). The resulting errors in the surface heat flux $H_{Q_0}$ and surface buoyancy flux $B_0$ depend on the meteorological forcing. From the yearly averaged meteorological data, the errors are $\delta_{H_{Q0}} = 6.1$ W m$^{-2}$ and $\delta_{B_0} = 2.1 \times 10^{-9}$ W kg$^{-1}$, which is 5 % of the yearly averaged $H_{Q_0}$ and $B_0$. We

have now included the error on the heat fluxes in lines 164-165: "*The uncertainty in wind speed and relative humidity estimates leads to an average uncertainty of 5 % and 3 % of the surface heat fluxes (Sect. 2.2), respectively.*"

[Figure]

Figure R1.1: One month-long time series of wind speed measured by the Lucerne (LUZ) and Rotsee (ROT) stations, and estimated from Neural Network Fitting (NNF) from November to December 2020.

[Figure]

Figure R1.2: Box plots of the wind speed measured at the Lucerne (LUZ) and Rotsee (ROT) stations, and estimated from Neural Network Fitting (NNF). The box plots are based on the dataset of Fig. R1.1.

**References:**

Fink, G., Schmid, M., Wahl, B., Wolf, T., and Wüest, A.: Heat flux modifications related to climate-induced warming of large European lakes, Water Resour. Res., 50, 2072–2085, https://doi.org/10.1002/2013WR014448, 2014.

McJannet, D. L., Webster, I. T., and Cook, F. J.: An area-dependent wind function for estimating open water evaporation using land-based meteorological data, Environ. Modell. Software, 31, 76–83, https://doi.org/10.1016/j.envsoft.2011.11.017, 2012.

Philippopoulos, K. and Deligiorgi, D.: Application of artificial neural networks for the spatial estimation of wind speed in a coastal region with complex topography, Renewable Energy, 38, 75–82, https://doi.org/10.1016/j.renene.2011.07.007, 2012.

**RC3:** *Methods 2.2.*

*L 160-161. Is the SW measured or parametrized? From section 2.1 it seems that it is measures but then in 2.2 it seems to be parametrized. Being a widely available parameter, I do not see the need to compute it.*

**AR3:** Incoming solar radiation reaching the lake surface ($R$) is directly measured (Sect. 2.1, line 152). However, a part of $R$ is reflected at the lake surface and is not included in the shortwave radiation entering the lake $H_{SW,0}$ (Sect. 2.2, lines 175-176). To compute $H_{SW,0}$, the albedo of direct and diffuse solar radiation is parametrized as a function of the cloudiness (Fink et al., 2014).

**Reference:**

Fink, G., Schmid, M., Wahl, B., Wolf, T., and Wüest, A.: Heat flux modifications related to climate-induced warming of large European lakes, Water Resour. Res., 50, 2072–2085, https://doi.org/10.1002/2013WR014448, 2014.

**RC4:** *L172. Are there measurements to support the S value? Did you perform any sensitivity to assure that possible variations had no effect on your evaluations at a seasonal scale?*

**AR4:** Yes, we have salinity estimates from conductivity profiles collected over the year. The surface salinity increases from summer to winter by approximately $\Delta S \approx 0.5$ g kg$^{-1}$ due to vertical mixing between the epilimnion and hypolimnion. The associated change of density is around $\Delta_S \rho \approx 0.4$ kg m$^{-3}$, which is almost one order of magnitude lower than the seasonal change of density due to surface temperature $\Delta_S \rho \approx 3$ kg m$^{-3}$. We added the following sentence in lines 187-189: "*Our analysis assumed a constant salinity over the year since the seasonal changes in surface density are controlled by temperature fluctuations rather than variations in salinity.*"

**RC5:** *L176. From this paragraph it seems that HQ0 is always a negative loose term, while the shortwave is the only one term that contributes to heating. On the contrary LWin and HC can be positive too. In general the way to manage the signs of the fluxes in this section terms is a bit confusing. I suggest to reason in term of H0net and B0net only (1+2 eq), without distinguishing between SW and other terms. What is important to verify is whether the net flux is positive or negative. This suggestion should be extended to the other sections of the paper.*

**AR5:** All the heat fluxes are defined positive in the upward direction (cooling), as explained in lines 170-171 of the revised manuscript. Even if some of the surface heat fluxes can be negative (heating) as mentioned by the reviewer, the total surface heat flux $H_{Q_0}$ remains indeed positive for most days. This continuous surface cooling is mainly due to the loss of longwave radiation. We believe that the confusion comes from lines 175-176 of the submitted manuscript where we opposed surface cooling to radiative heating. We modified this sentence as follows (now lines 192-193): "*The net buoyancy flux at the surface is $B_{0,net} = B_0 + B_{SW,0}$. $B_{0,net} > 0$ indicates a destabilizing buoyancy flux (net cooling) whereas $B_{0,net} < 0$ indicates a stabilizing buoyancy flux (net heating).*"

We agree with the reviewer regarding the use of the net heat and buoyancy fluxes. The cooling and heating phases must be determined from $H_{0,net}$, which is directly related to $B_{0,net}$. However, the driving force of TS is surface cooling, expressed by a destabilising surface buoyancy flux $B_0$. Distinguishing $B_0$ from the radiative (penetrative) buoyancy flux is required to determine the convective velocity scale (Fig. 4) and the transport scaling formulae (Sect. 2.6). The use of $B_{0,net}$ and $B_0$ is already explained in lines 193-194 of the revised manuscript.

**RC6:** *Methods 2.4. In the methods aimed at detaching the thermal syphons I would have expected to see the vertical component of the velocity as a target variable, in particular before the beginning of the event. Is there any reasons why you did not mention it? Given the uncertainty on the wind data, I think it could be a better way to distinguish between thermal syphons and wind driven flow (see e.g. Fer et al. 2002).*

**AR6:** Thank you for the suggestion. The vertical velocity measured by the ADCP has indeed a different signature between TS events and wind-driven flows and we initially tried to use it in the "wind filter" of our algorithm. TS events are characterized by convective plumes with alternating upward and downward vertical velocities (Fig. 3c), whereas wind-driven cross-shore flows are associated with strong downwelling at MT, when they are directed in the x-direction ($U_x > 0$). The challenging aspect of this approach is the definition of the criterion used to distinguish the two different signatures in the vertical velocity. An option is to use a threshold value for $U_z$ (which is larger for downwelling than convection) or to focus on the change of sign of $U_z$ over a certain period. Regarding reproducibility, these criteria involve arbitrary threshold values that are not necessarily physically grounded and could be more system-dependent than our criterion based on the Monin-Obukhov length $L_{MO}$. We also think that vertical velocities can be more difficult to measure in the field than $L_{MO}$, as they require high resolution ADCP data. Additionally, the wind filter based on $L_{MO}$ can be applied a priori to predict the occurrence of TS on a specific day. Although there is uncertainty in the wind speed estimation, our wind filter correctly discarded all the cross-shore flows associated with downwelling.

We are not sure what the reviewer refers to in Fer et al. (2002). To our understanding, Fer et al. (2002) did not use the vertical velocity to distinguish between TS and wind-driven flows. They used the ratio $z/L_{MO}$ to study the effects of wind on TS (with $z$ the depth of interest, see their figure 7), which is similar to our approach. Yet, we still think that using vertical velocities could be an interesting approach to try in the future and we mentioned it in line 684-686 in Appendix D: "*Additional filters could be implemented to distinguish between TS and wind-driven cross-shore flows, based for example on high-resolution vertical velocity measurements (…).*"

**Reference:**

Fer, I., Lemmin, U. and Thorpe, S. A.: Winter cascading of cold water in Lake Geneva, J. Geophys. Res., 107(C6), 3060, https://doi.org/10.1029/2001JC000828, 2002.

**RC7:** *Table1. Separate the extremes of the range with a "-" in place of a ",". Why no range for Ux and Uz?*

**AR7:** We initially used "," to avoid confusion with the minus sign but we have now followed the reviewer's suggestion and replaced "," by "-" for the ranges in Table 1, with brackets around negative values. We also added the range of values for $U_x$ and $U_z$ in Table 1.

**RC8:** *Figure 3,4,7,8. The figures of this paper are too much dense of information. The effort to make them fully informative has the counter-effect to confuse the reader with too much data and is not efficient in highlighting a clear message. Make an effort to make the figures clearer, with less but more direct information, and eventually reduce the number of figures. ( Fig. 8 in particular is really hard to follow)*

**AR8:** We appreciate the critical assessment of the figures provided by the reviewer. We acknowledge that our figures provide a lot of information but we tried to provide a detailed description in each caption. The two other reviewers are both very positive about the figures (R2: "*I (...) found all of the figures engaging*", R3: "*The writing and presentation are mostly very good.*"). We still tried to improve the clarity of Fig. 3, 4, 7 and 8 as follows:

- Figure 3: see the specific answer AR9 below.
- Figure 4: we removed all the error bars in Fig. 4a to improve the readability and removed the corresponding paragraph (lines 314-317 of the submitted manuscript).
- Figure 7: the months are now clearly indicated for each row of subpanels, which helps to better understand the organization of the figure.
- Figure 8: we moved Fig. 8b to Appendix B (now Fig. B1) as it was not directly related to the seasonality of the time scales. We replaced it by a schematic illustrating the different time scales over the cooling phase, as suggested by Referee #2 (see answer AR2 to Referee #2).

**RC9:** *Fig. 3. The first panel is useless. In the second panel limit the plot of B0net. Velocity and temperature contours together limit the readability. Look at Fig. 3 of Fer et al. 2002 as an example of a good representation of a single event: a single line of Ux and Uz is much clearer. Finally the contours of temperature between 13.00 and 17:00 looks like affected by an error in the interpolation. Do you have enough thermistors? If so, how do you explain the different pattern? How do you explain the rapid changes in signs of Uz? Please comment.*

**AR9:**

- Fig. 3a: We do not think that this panel is useless, as it illustrates the seasonality of the diurnal cycle shown for a specific day in the three other panels. In particular, Fig. 3a shows the seasonality of (1) the duration of the heating and cooling phases and (2) the magnitude of the forcing. The seasonality of the diurnal cycle is a key aspect to understand how the occurrence (Sect. 4.2) and the flushing period (Sect. 4.4) vary over the year. Since the role of Fig. 3a was maybe unclear, we added several references to this panel in the text (see lines 322, 330 and 397). We also think that the reviewer's comment comes from the fact that Fig. 3a seemed disconnected from the three other panels. We have now better linked them by clearly indicating with the dashed arrow in Fig. 3a where the diurnal cycle of Fig. 3b-d appears.
- Fig. 3b: Showing only $B_{0,net}$ does not provide enough information in our view. $B_{0,net}$ implicitly indicates the cooling and heating phases, associated with destabilising and stabilising surface buoyancy fluxes, respectively. However, $B_0$ and $B_{SW,0}$ are relevant to demonstrate that the main driver of the diurnal cycle is the solar radiation, and not the temporal change of surface cooling ($B_0$ remains positive all day), as explained in lines 274-277. To better emphasize $B_{0,net}$ in Fig.

3b, we have increased the thickness of its line and we have represented $B_0$ and $B_{SW,0}$ with dotted lines.

- Fig. 3c-d: The goal of this figure is not only to identify a TS event, as in Fig. 3e of Fer et al. (2002b), but also to provide the main characteristics of the convective circulation (opposite cross-shore flows, thickness of TS, region of maximum velocity) and to indicate the region where the transport is calculated (blue curves in Fig. 3d). The latter requires to show velocity contours, and not only depth-averaged velocities. This is similar to the event presented in Fig. 2 of Fer et al. (2002a). Note that Fig. 3e of Fer et al. (2002b) focuses on the cross-shore velocity $U_x$ and the along-shore velocity $U_y$. It does not include the vertical velocity $U_z$ as in our Fig. 3c.

  We also want to keep the isotherms in Fig. 3d, as they show the stratification induced by the density current. To improve the readability, we have increased the spacing between the isotherms to 0.05 °C and we show them in gray. The dense surface temperature contours during the second day are due to the strong surface heating captured by the surface thermistor. The temperature has been linearly interpolated between the thermistors and there is no thermistor between the surface and 3 m depth due to rowing restrictions (lines 137-138 and Table A1), which leads to dense isotherms down to 2 m depth (2 m height above the bottom). We have now extended the upper limit of the y-axis to 4 m height to show the location of the surface thermistor. We are also mentioning the linear interpolation of temperature in the caption, line 307: "*Black lines are 0.05 °C-spaced isotherms that are linearly interpolated between each thermistor*". To gain clarity, we replaced the name of the three periods in Fig. 3d by numbers and explained them in the caption (lines 309-310).

  Vertical velocities in Fig. 3c have a temporal resolution of 15 min, as explained in the figure caption. We did not interpolate them over time: one value is shown at each depth every 15 min. The "rapid" changes in signs of $U_z$ come from the upward-downward motion of the convective plumes every 15 min. The duration of half of a convective overturn can be estimated as $\tau_{conv} = h_{MT}/U_z$ with $h_{MT} \approx 4$ m the depth at MT and $U_z \approx 0.005$ m s$^{-1}$ the velocity of convective plumes. This gives $\tau_{conv} \approx 13$ min, which is less than the temporal resolution of Fig. 3c. The convective plumes can thus have opposite directions between two consecutive measurements.

**References:**

Fer, I., Lemmin, U. and Thorpe, S. A.: Contribution of entrainment and vertical plumes to the winter cascading of cold shelf waters in a deep lake, Limnol. Oceanogr., 47(2), 576–580, https://doi.org/10.4319/lo.2002.47.2.0576, 2002a.

Fer, I., Lemmin, U. and Thorpe, S. A.: Winter cascading of cold water in Lake Geneva, J. Geophys. Res., 107(C6), 3060, https://doi.org/10.1029/2001JC000828, 2002b.

**RC10:** *Results 3.4. The R2 values are really low, I would limit the analysis only to the variables which show at least a trend (not the case of tauf for example). In the conclusion you defined "robust" these relationships , but these R2 do not support these conclusions. I would be more cautious to base the conclusions on the basis of these results.*

**AR10:** We acknowledge that the $R^2$ values are low, which is not surprising knowing the natural variability of the process. We attributed the strong variability of $q_{avg}$ and $\tau_F$ to the fluctuations of the thickness of TS, as mentioned in Sect. 3.4 (lines 383-384) and discussed in Appendix D (lines 699-707). Despite the scatter between days, the four variables of Fig. 6, including $\tau_F$, show a linear trend. This is confirmed by the low p-value ($p_{val,F} < 10^{-53}$ for the four variables), which indicates that the slopes of the linear fits are significantly different from zero. We agree with the reviewer that the term "robust" was excessive. We removed it from the conclusion, lines 602-603 now read: "*This study provides a field validation of this laboratory and theoretically based scaling.*"

**RC11:** *References. A careful review of the references is needed (for example Rao and Schwab, Meyers and Dale are not present)*

**AR11:** Thank you for the comment, we apologize for this issue. We realized that our reference management software did not work properly. We have added the following missing references:

- Lines 755-756: Fink, G., Schmid, M., Wahl, B., Wolf, T., and Wüest, A.: Heat flux modifications related to climate-induced warming of large European lakes, Water Resour. Res., 50, 2072–2085, https://doi.org/10.1002/2013WR014448, 2014.
- Lines 787-789: McJannet, D. L., Webster, I. T., and Cook, F. J.: An area-dependent wind function for estimating open water evaporation using land-based meteorological data, Environ. Modell. Software, 31, 76–83, https://doi.org/10.1016/j.envsoft.2011.11.017, 2012.
- Lines 790-791: Meyers, T. and Dale, R.: Predicting daily insolation with hourly cloud height and coverage, J. Climate Appl. Meteor., 22, 537–545, https://doi.org/10.1175/1520-0450(1983)022<0537:PDIWHC>2.0.CO;2, 1983.
- Lines 802-803: Rao, Y. R. and Schwab, D. J.: Transport and mixing between the coastal and offshore waters in the Great Lakes: a review, J. Great Lakes Res., 33, 202–218, https://doi.org/10.3394/0380-1330(2007)33[202:TAMBTC]2.0.CO;2, 2007.

**Response to Referee #2**

**R (Referee):**

*HESS three Prinicipal Review Criteria*

*1. Scientific significance is excellent with new concepts, methods, and data.*

*2. Scientific quality is good. The approach and applied methods are valid and the results are appropriately discussed.*

*3. Presentations quality is good. There is an appropriate number and quality of figures/tables, appropriate use of English language. I found that there were few typos and that the English is good. The use of some symbols and abbreviations is confusing.*

*General Comments*

*The preprint addresses an important aspect of differential cooling with new concepts and data. With an extensive data set the thermal siphon process is shown to flush the near shore region. A simple model based on practically measured or available data is used to predict this process and it's seasonal variability. I enjoyed reading all five sections and the appendices and found all of the figures engaging. In summary I beleive the work represents a significant contribution to the field and is well suited to HESS.*

**A (Authors):** We are grateful to Referee #2 for his/her positive feedback and for his/her comments about the clarity of the manuscript. We address them below.

*My only general comment is that the clarity of the paper's main findings are obscured somewhat by the complex collection of abbreviations and symbols. In the first three specific comments below I address this and other clarity issues that I think should be addressed.*

*Specific comments*

*Three specific comments related to the overall clarity:*

**RC1 (Referee's Comment #1):**
*The along-x locations and their labels are confusing even after the reader is comfortable with the XZ description of the lake:*

*L_lit is the distance from one end of the lake along the thalweg to MT. So MT was located at the location where the photic zone reaches the bottom? I don't think this is ever stated, rather it seems MT is located at an arbitrarily shallow location along the thalweg.*

*L_ML the distance from the same origin along the thalweg to the isobath that matches the depth of the mixed at MB.*

*x_q the distance from the same origin to MT (where q is measured/predicted but not clearly linked between Figures 1 and 2).*

*l_p the length of the plateau (not indicated in Figure 2).*

*I think the formatting of these labels should be more consistent (e.g. a capital letter L followed by a subscript) and that x_q or L_lit be omitted. A similar simplification would help with the depths (d_p,d_MT,h_TS,h_lit, etc). I never could figure out what MT and MB stood for.*

**AR1 (Authors Response #1):** Regarding the notations, we have followed the reviewer's suggestion and now used a capital letter $L$ for distances along $x$ ($L_{lit}$, $L_{ML}$, $L_p$) and a lower case $h$ for distances along $z$ ($h_p$, $h_{MT}$, $h_{TS}$, $h_{lit}$, $h_{ML}$). The letter B in MB stands for "Background". The letter T in MT stands for "Thermal siphon". We added these definitions to lines 130-132 of the revised manuscript: "*We monitored the background stratification at the deepest location ("background mooring" MB, approx. 16 m deep) as well as the dynamics of TS offshore from the plateau region ("TS mooring" MT, approx. 4 m deep), from March 2019 to March 2020 (Fig. 2a).*"

Regarding the specific length scales:

- Indeed, $L_{lit}$ is the length of the littoral region flushed by TS at MT. As discussed in Sect. 4.3 (lines 529-531), this length scale depends on the location of measurements along $x$ because it is used to parametrize the $x$-dependent flushing time scale.

  The location of MT is constant over the entire year, and we never stated in the manuscript that it was selected on a photic-depth criterion (which is seasonally dependent). The photic zone, estimated from repeated Secchi Depth measurements, was deeper than the water depth at MT except during the productive period in late summer 2019, where the photic depth was reduced to ~ 4 m.

  MT was positioned along the thalweg, with the two following criteria: (1) to be in the sloping region and (2) to be shallower than the mixed layer depth during cold summer nights ($h_{ML} \approx 5$ m). These two conditions allowed us to capture downslope TS already in summer.

  We are now better explaining the location of MT in lines 132-133: "*Mooring MT was located along the thalweg, at the beginning of the sloping region. This shallow water column is already vertically mixed in summer by the action of surface cooling.*"

- $L_{ML}$ is defined from the mixed layer depth because it corresponds to the distance over which differential cooling takes place (Sect. 4.3, lines 526-528).

- We used $x_q$ to refer to the location of discharge measurements in other studies (Table 2). In our case, the length of the littoral region $L_{lit}$ is equal to $x_q$. We agree that these different notations might be confusing for the reader and we have now replaced $x_q$ by $L_{lit}$ in Table 2. We specify in the caption that $L_{lit}$ is defined based on the location of discharge measurement: "*The littoral region of length $L_{lit}$ and average depth $h_{lit}$ is defined as the region upslope of the location of discharge measurement.*"

- We have added the $x$-axis, $L_p$ and $L_{lit}$ in Figure 2.

**RC2:** *Although the transect data in the schematic represents an efficient use of space and looks great I think it unnecessarily complicates the schematic. The schematic should address the seasonal cycle, identify the plateau, perhaps include the equation q=c_q h (BL)^1/3 or similar equation for U, and serve as a road map or foreshadowing for the rest of the paper. Something like Table 2 added to the introduction could compliment the schematic. Where is the origin x=0 on the map in Figure 1? Why not identify the plateau in the schematic? Could the authors incorporate a graphic illustrating the essential time scales? If aspects at the end of the paper are too complicated to include in the initial schematic provide a revised schematic at the end of the results or in the discussion. I recognize the authors have spent some time linking the text and figures including Figure 1 and Table 1 together but it still needs improvement.*

**AR2:** Thank you for helping us improve Fig.1. We modified the figure based on the comments of the reviewer.

[Figure]

**Figure 1: Data-based schematic of the cooling-driven thermal siphon. The schematic shows the plateau, littoral and mixed regions, the seasonality of the forcing and the variables used for the transport scaling. Here, the littoral zone is the region upslope of MT, where the current velocity is measured and transport variables are calculated. The cross-shore temperature field is linearly interpolated from a transect of CTD (Conductivity-Temperature-Depth) profiles collected in the morning on 22 August 2019 (08:20–08:50 UTC), from x = 225 m (white dashed line) to x = 714 m. Black dots on the x-axis show the location of the profiles. The green dashed line in the seasonality diagram corresponds to the transition period between the mixing period (winter) and the stratified period (summer), when the mixed layer is not well defined.**

We want to keep the transect data in Fig. 1, as it provides important information on (1) differential cooling and (2) TS-induced stratification. However, we understand the concern of the reviewer about the clarity of the figure. To simplify the schematic, we kept the colormap but removed the isotherms and replaced the vertical dashed lines with points on the $x$-axis.

As suggested by the reviewer, we have now added the plateau to the schematic and indicated $L_p$ and $h_p$. We are using four boxes on top of the schematic to list the scales related to the littoral region, the plateau region and the thermal siphon, and to address the seasonal cycle with a conceptual graphic showing the seasonality of the forcing conditions ($B_0$, $h_{ML}$, $L_{ML}$). We have also included the scaling formulae for $U$, $q$ and $\tau_F$ (Eqs. (9), (10), (11)) in the box about TS.

The origin $x = 0$ was already indicated in Fig.1. We believe that the reviewer is referring to the bathymetric map of Fig. 2. We have added the x-axis and its origin in Fig. 2 (see AR1).

In the submitted version of the manuscript, the time scales were introduced in different sections: Sect. 3.2 ($\tau_c$), Sect. 3.5 ($\tau_t$) and Sect. 4.2 ($\tau_{ini}, \tau_{mix}$). We realized that this could be confusing for the reader and it might explain why the reviewer is asking for an overview of the time scales in Fig. 1. We are now introducing $\tau_c$, $\tau_{ini}$ and $\tau_t$ in Sect. 2.6 (lines 253-261), as they are based on previous studies. We have kept $\tau_{mix}$ in Sect. 4.2 as it is a modification of $\tau_{ini}$ that we propose in this study. It is difficult to include all these time scales in Fig. 1 since the schematic shows the spatial and not the temporal variability of TS. We have included the initiation time scale in the box about TS in Fig. 1 and we have added a schematic illustrating $\tau_c$, $\tau_{ini}$, $\tau_{mix}$ and $\tau_t$ in Fig. 8a, once these time scales have been introduced in the text.

We hope that the different changes mentioned above help to better link the text to the figures.

**RC3:**

*The language related to flow direction is sometimes confusing. I think this is partially due to the fact that the shorelines to the northwest and southeast are closer to both stations than the shore line to the northeast. The authors should explicitly state early in the paper and repeat in several captions that offshore flow is southwestward flow or something similar, line 191 is inadequate. I don't think the authors ever comment on along shore flow, tell the reader why it's ignored or if there's none.*

**AR3:** Thank you for this comment. We should have better introduced the framework that we are using regarding the flow direction. Our motivation in this study is to quantify the flushing of the littoral plateau region at the north-eastern end of the lake. We use a 2D framework to study the convective circulation (Fig. 1) and focus on the TS-induced transport along the thalweg (x-axis). The x-axis defines what we call the "offshore direction". Due to its elongated shape, Rotsee is suitable for this 2D framework, with the strongest TS flowing preferentially along the thalweg. The along-shore flow (y-axis) is generally small but not necessarily zero as TS can be slightly deviated from the x-axis by Coriolis or topographic effects. The limitations of the 2D framework are now discussed in Sect. 4.5, in lines 570-572: *"The 2D framework of TS requires specific validation in more complex nearshore systems and large lakes, where the topography, large-scale circulation and Coriolis may also affect the TS dynamics (Fer et al., 2002b). In these systems, the along-shore velocity component of TS must be considered in the cross-shore transport analysis."* The 3D aspects of TS are out of the scope of this study but could be investigated with 3D numerical experiments, as mentioned in lines 583-584: *"The effects of more complex bathymetries, departing from our 2D framework, could be further investigated with 3D numerical simulations."*

We are now explaining our 2D framework and clarifying the language about the flow direction in Sect. 2.1, by adding the following paragraph in lines 126-129: *"This study focuses on quantifying TS originating from the north-eastern plateau region (Fig. 2a). Because of the elongated shape of Rotsee, we use the 2D (x, z) framework shown in Fig. 1 by orienting the x-axis along the thalweg. We assume that TS flows preferentially along the x-axis, and we neglect the influence of perpendicular flows. We will now refer to the north-eastern end of the lake as the "shore" and call the direction of the x-axis the "offshore direction"."*

In addition, we also modified lines 206-208 to define the cross-shore velocity: *"The horizontal velocity was projected onto the x-axis (angle of 56° from north), which crosses the isobath at MT perpendicularly (Fig. 2a). Following the 2D framework of Fig. 1, we will now call $U_x$ the "cross-shore velocity"."* We are specifying "south-westward flows" when we refer to cross-shore flows in the captions of Figs. 3 and 5.

*Other specific comments*

**RC4:** *I was expecting to see more transects demonstrating the TS during other times of the year e.g. TS in July, October and December, were there no others collected?*

**AR4:** Transects of temperature profiles were collected during twelve campaigns from August to December 2019. We used one transect in Fig. 1 to show the cross-shore temperature distribution. We did not include the other transects in the manuscript because they do not clearly show the seasonality of TS. The bottom stratification and the TS thickness are similar between transects. The main seasonal differences are the depth and length of the mixed littoral region ($L_{ML}, h_{ML}$) and the duration of the TS events. Temperature transects are more relevant for studying the short-term variability of TS over one diurnal cycle (periods shown in Fig. 3d), which is not the objective of this study.

We believe that the reviewer expected to see more transects because we mentioned the different campaigns in lines 133-135 of the submitted manuscript. We have simplified this paragraph to only refer to the transect on August 22 (now lines 166-167): "*We captured the spatial variability of TS in the morning on 22 August 2019 (Fig. 1) by collecting 11 Conductivity-Temperature-Depth (CTD) profiles (Sea&Sun CTD 60M) along the x-axis, between 2 and 14 m depth (Fig. 2a).*"

**RC5:** *Figure 2. Provide the depth at MT in Figure 2 (b). I think the map of Switzerland should be idienfied as a map of Switzerland.*

**AR5:** The schematic of the mooring in Fig. 2b refers to both MB and MT. We have indicated the depth of both moorings on the map of Fig. 2a. We are mentioning the map of Switzerland in the caption: "*The location of Rotsee is shown on the map of Switzerland with a black dot.*"

**RC6:** *Lines 145 to 152 - has anyone ever done this before for winds or humidity? explain why you think the simpler approach failed. Can you provide a separate R^2 for the northerly and westerly wind components, or the along and across axis wind components?*

**AR6:** Artificial Neural Networks are commonly used for the spatial interpolation of meteorological parameters, including wind speed (Öztopal, 2006; Kusiak and Li, 2010; Philippopoulos and Deligiorgi, 2012) and relative humidity (Yasar et al., 2012; Philippopoulos et al., 2015). Unlike pressure, air temperature and solar radiation, a simple linear interpolation cannot be used for wind speed and relative humidity because these two parameters are highly variable over time and are dependent on the surrounding environment. Philippopoulos and Deligiorgi (2012) showed for instance that Neural Networks are more performant than traditional interpolation methods of wind speed.

The values of $R^2$ and $E_{RMS}$ for cross-shore (x-axis) and along-shore (y-axis) wind components are $R_x{}^2 \approx 0.85$, $E_{RMS,x} \approx 0.61$ m s$^{-1}$ and $R_y{}^2 \approx 0.64$, $E_{RMS,y} \approx 0.26$ m s$^{-1}$, respectively. We did not include this information in the manuscript, as we did not use the wind direction in the analysis.

**References:**

Kusiak, A. and Li, W.: Estimation of wind speed: A data-driven approach, J. Wind Eng. Ind. Aerodyn., 98, 559–567, https://doi.org/10.1016/j.jweia.2010.04.010, 2010.

Öztopal, A.: Artificial neural network approach to spatial estimation of wind velocity data, Energy Convers. Manage., 47, 395–406, https://doi.org/10.1016/j.enconman.2005.05.009, 2006.

Philippopoulos, K. and Deligiorgi, D.: Application of artificial neural networks for the spatial estimation of wind speed in a coastal region with complex topography, Renewable Energy, 38, 75–82, https://doi.org/10.1016/j.renene.2011.07.007, 2012.

Philippopoulos, K., Deligiorgi, D., and Kouroupetroglou, G.: Artificial Neural Network modeling of relative humidity and air temperature spatial and temporal distributions over complex terrains, in: Pattern Recognition Applications and Methods, vol. 318, edited by: Fred, A. and De Marsico, M., Springer International Publishing, Cham, 171–187, https://doi.org/10.1007/978-3-319-12610-4_11, 2015.

Yasar, A., Simsek, E., Bilgili, M., Yucel, A., and Ilhan, I.: Estimation of relative humidity based on artificial neural network approach in the Aegean Region of Turkey, Meteorol. Atmos. Phys., 115, 81–87, https://doi.org/10.1007/s00703-011-0168-2, 2012.

**RC7:** *Line 239 I think this is ok for B_0 and is discussed later but I'm not so sure about L_ML, wouldn't this often increase over the cooling period?*

**AR7:** Yes, the mixed layer can deepen by ~3 meters during intense daily cooling periods in late summer, which leads to an increase of $L_{ML}$ of ~70 meters over the same periods. This increase is more limited in late autumn, due to the weaker convection. We are averaging $L_{ML}$ over the cooling phase since we are interested in the estimation of daily averaged transport variables only. We are not investigating here the short-term temporal changes of $U$ and $q$ over the cooling phase. Moreover, the daily increase of $L_{ML}$ changes the velocity scale $(B_0 L_{ML})^{1/3}$ by $O(10^{-3})$ m s$^{-1}$, which is one order of magnitude lower than $(B_0 L_{ML})^{1/3}$. For typical summer conditions with $B_0 \sim 10^{-7}$ W kg$^{-1}$ and $L_{ML} \sim 200$ m for example, an increase of $L_{ML}$ by 70 meters changes the velocity scale $(B_0 L_{ML})^{1/3}$ by ~0.003 m s$^{-1}$.

**RC8:** *Table 1 would benefit from some recomposition to aid in connecting the four columns, particularly the fourth column, e.g. swap the third and fourth column and justify the 'definition and equation' column left.*

**AR8:** We have followed the reviewer's suggestion. We are now providing the ranges of values in the third column and the equations in the fourth column (justified to the left).

*Technical corrections*

**RC9:** *Whether limnology is patriarchal or not the reference to 'fathers of limnology' reads a little too patriarchal.*

**AR9:** We have replaced "fathers of limnology" by "pioneer limnologists" (line 35).

**RC10:** *The whole sentence beginning 'Such shift' on line 594 needs improvement, to start, change 'Such shift' to 'Such a shift'.*

**AR10:** We have modified this sentence as follows (now lines 607-609): "*Such a timing has implications for the transport of dissolved compounds, with, for instance, stronger exchange between littoral and pelagic waters at a time of high primary production (summer and daytime).*"

**RC11***: line 373 and 374 change shadow to shading.*

**AR11:** We have replaced "blue shadow" by "blue shading" in line 396 and removed the reference to "red shadow".

**RC12:** *line 376 refer to figure 7 for the histograms.*

**AR12:** We have added the reference to Fig. 7b, d, f, h, j (now line 398).

**RC13:** *line 475 remind the reader what the depth is at MT.*

**AR13:** The depth of MT is now specified in the caption of Fig. 8 (now line 469): "*The mixed layer depth is expressed as the relative depth $h'_{ML} = h_{ML} - h_{MT}$, with respect to $h_{MT} = 4\ m$.*"

**Response to Referee #3**

**R (Referee):**

*In this manuscript, the authors use a unique year long time series to estimate the frequency and strength of the thermal siphon in lakes and the influence on flushing of the littoral zone. This is an interesting dataset and addresses an important concept in physical limnology. The writing and presentation are mostly very good.*

**A (Authors):** We thank Referee #3 for his/her encouragement and his/her interesting questions about the interpretation of the results. We address all the comments from Referee #3 below.

*I have a few general comments that I would like to see addressed though:*

**RC1 (Referee's Comment #1):** *What is the magnitude of the outflow near the study site? Is it important relative to magnitude of the flushing rates?*

**AR1 (Authors Response #1):** There is no in-situ measurement in the outflow but the discharge can be estimated from the simulations of the Swiss river network (dataset MQ-GWN-CH from the Federal Office for the Environment). The monthly averaged simulated discharges of the Rotsee outflow in $m^3$ $s^{-1}$ are available here:

https://api.geo.admin.ch/rest/services/ech/MapServer/ch.bafu.mittlere-abfluesse/67200/extendedHtmlPopup?lang=en.

The average discharge is $\sim 0.1$ $m^3$ $s^{-1}$, which corresponds to a specific-width discharge at MT (total width of 150 m) of $q_{out} \sim 7 \times 10^{-4}$ $m^2$ $s^{-1}$. This estimate is more than one order of magnitude smaller than the TS discharge $q_{avg} \sim 10^{-2}$ $m^2$ $s^{-1}$ (Fig. 6). The effects of the outflow on the TS dynamics are thus negligible. We are now mentioning the low discharge of the inflow and outflow on lines 112-113: *"The main in- and outflows are located at the south-western and north-eastern ends of the lake, respectively, and have a low discharge of $\sim 0.1$ $m^3$ $s^{-1}$."*

**RC2:** *What influence does the three-dimensionality of the littoral zone play? From my understanding, the entire framework here is 2D, but how uniform do you suppose q is across the lake? What are the limits of your results for other lakes in that context? It appears that Rotsee is about 2D as it gets, but is there a littoral zone aspect ratio where this all falls apart?*

**AR2:** The reviewer is correct: we used a 2D framework in this study and we focused on the lateral transport along the x-axis, as our objective was to quantify the flushing of the littoral plateau region at the north-eastern end of the lake. The elongated shape of Rotsee is suitable for this 2D framework, which is similar to the nearly 2D sidearm circulation observed in reservoirs (Adams and Wells, 1984; Monismith et al., 1990). We decided to take our measurements along the lake thalweg, which is the preferential direction of TS according to 3D numerical simulations (Ramón et al., 2019). We expect $q$ to be lower if it is measured away from the thalweg, closer to the north-eastern or south-western shores. The validity of such a 2D framework has to be further evaluated in more complex nearshore systems that might deviate from the conceptual model adopted here. Coriolis effect and local bathymetry perturbations might have to be included in these cases. For instance, the downslope flows observed by Fer et al. (2002) in Lake Geneva are not perpendicular to the shore due to spatial irregularities of the littoral region. The 3D aspects of TS are out of the scope of this study, but could be a motivation for future work. In particular, the effect of the littoral zone aspect ratio on the TS dynamics could be investigated with 3D numerical simulations, as mentioned in lines 583-584 of the revised manuscript: *"The effects of more complex bathymetries, departing from our 2D framework, could be further investigated with 3D numerical simulations."*

We have added a paragraph in lines 126-129 to explain our 2D framework. We refer to the answer AR3 to Referee #2 for more details about this paragraph. We have also added a few sentences about the 3D effects in other lakes in lines 570-572: *"The 2D framework of TS requires specific validation in more complex nearshore systems and large lakes, where the topography, large-scale circulation and Coriolis may also affect the TS dynamics (Fer et al., 2002b). In these systems, the along-shore velocity component of TS must be considered in the cross-shore transport analysis."*

**References:**

Adams, E. E. and Wells, S. A.: Field measurements on side arms of Lake Anna, Va., J. Hydraul. Eng., 110, 773–793, https://doi.org/10.1061/(ASCE)0733-9429(1984)110:6(773), 1984.

Fer, I., Lemmin, U., and Thorpe, S. A.: Winter cascading of cold water in Lake Geneva, J. Geophys. Res., 107, 3060, https://doi.org/10.1029/2001JC000828, 2002.

Monismith, S. G., Imberger, J., and Morison, M. L.: Convective motions in the sidearm of a small reservoir, Limnol. Oceanogr., 35, 1676–1702, https://doi.org/10.4319/lo.1990.35.8.1676, 1990.

Ramón, C., Doda, T., Ulloa, H., and Bouffard, D.: Density currents induced by night-time cooling: offshore transport of littoral waters, in: Geophysical Research Abstracts, Vienna, Austria, 7-12 April 2019, EGU2019-970, 2019.

**RC3:** *Given the time of year where TS is most prevalent, does this play a role in accelerating autumn turnover?*

**AR3:** This is an interesting point. Yet, the effects on autumn turnover are out of the scope of our study and will not be discussed in the manuscript. The intrusion of TS at the base of the mixed layer modifies the vertical thermal structure by bringing cold water above the thermocline. This advective heat flux enhances the cooling of the surface layer and should accelerate the deepening of the mixed layer in autumn, as observed for differential heating under ice (Ulloa et al., 2019). The shear induced by the intrusion of TS might also increase vertical mixing and lead to a faster erosion of the stratification (Strang and Fernando, 2001). However, the stratification induced by TS can prevent the water column from becoming entirely mixed in winter. A steady mixed layer can remain if the volume flux provided by TS balances the volume flux of mixed layer deepening (Wells and Sherman, 2001). As a result, the time of complete overturn would be delayed by TS and would only occur when TS stops later in winter (i.e., when $\tau_t > \tau_c$). In the case of Rotsee, TS rarely reaches the lake center and we do not expect a basin-scale effect on the autumn turnover. The effect should be more pronounced in lakes with a larger ratio $A_S/A_D$, where $A_S$ and $A_D$ are the surface areas of the shallow and deep regions, respectively (Wells and Sherman, 2001).

**References:**

Ulloa, H. N., Winters, K. B., Wüest, A., and Bouffard, D.: Differential heating drives downslope flows that accelerate mixed-layer warming in ice-covered waters, Geophys. Res. Lett., 46, 13872–13882, https://doi.org/10.1029/2019GL085258, 2019.

Strang, E. J. and Fernando, H. J. S.: Entrainment and mixing in stratified shear flows, J. Fluid Mech., 428, 349–386, https://doi.org/10.1017/S0022112000002706, 2001.

Wells, M. G. and Sherman, B.: Stratification produced by surface cooling in lakes with significant shallow regions, Limnol. Oceanogr., 46, 1747–1759, https://doi.org/10.4319/lo.2001.46.7.1747, 2001.

**RC4:** *TS is stronger in the summer, but less frequent. As noted, the summer is when this physical process might have the most impact on ecological and biogeochemical processes. Is there a way to compare TS across the months in a more quantitative fashion, for example, the total volume of water flushed in each month?*

**AR4:** We did not include the flushed volume in the manuscript because its estimation depends on the definition of the flushing period, which is arbitrary (Sect. 2.5, lines 230-231 of the revised manuscript). It is still interesting to compare the seasonality of the daily flushed volume with the seasonality of the occurrence and intensity of TS. We have included the seasonality of the flushing in Appendix C. We estimated the daily averaged flushed volume by dividing the total volume of water flushed every month by the number of days with measurements (Fig. C1a). The seasonal trend is similar to the occurrence, which indicates that the increase of occurrence of TS from summer to autumn overcomes the weakening of the transport over the same period. The occurrence of TS is thereby the primary factor controlling the seasonality of the littoral flushing in Rotsee. Interestingly, the volume flushed by each TS event (Fig. C1b) does not seem to depend on the season. It remains around 1-1.7 times the volume of the littoral region for every month. Therefore, a single TS event flushes on average the littoral region more than once.

**RC5:** *Figure 3d - the contours on the upper right corner look more like an artefact of the contouring than anything that might possibly be real?*

**AR5:** These dense isotherms are due to the linear interpolation of the strong surface heating captured by the surface thermistor. We have extended the y-axis in Fig. 3d up to the surface thermistor (height of 4 m) to make this interpolation more obvious. We are also mentioning the linear interpolation in the caption of Fig. 3. We refer to the answer AR9 to Referee #1 for more details.

**RC6:** *Figure 6 - are you forcing the intercept here? From the equations that seems the case (an intercept of 0), but that doesn't look like the best fit line, at least for (b)*

**AR6:** Yes, we are forcing the intercept to be zero for the four quantities in Fig. 6. We agree that this approach does not provide the best fit but it follows the scaling formulae. The results of the linear fitting with non-zero intercept are:

$U_{avg} = 0.35 \cdot (B_0 L_{ML})^{1/3} - 8 \times 10^{-4}$ m s$^{-1}$,

$U_{max} = 1.22 \cdot (B_0 L_{ML})^{1/3} - 0.006$ m s$^{-1}$,

$q_{avg} = 0.28 \cdot h_{lit}(B_0 L_{ML})^{1/3} + 0.003$ m$^2$ s$^{-1}$ ,

and $\tau_F = 2.09 \cdot L_{lit}(B_0 L_{ML})^{-1/3} + 2.99$ h.

Note that those results are not significantly different from the scaling with a forced intercept. We are now mentioning the zero intercept in the caption of Fig. 6 (line 391): *"The equation of the linear regressions with zero intercept (...)"*, and in Sect. 3.4, lines 374 and 381.